# STAR-CONVEXITY IN NON-NEGATIVE MATRIX FACTORIZATION

## ABSTRACT

Non-negative matrix factorization (NMF) is a highly celebrated algorithm for matrix decomposition that guarantees non-negative factors. The underlying optimization problem is computationally intractable, yet in practice gradient descent based solvers often find good solutions. This gap between computational hardness and practical success mirrors recent observations in deep learning, where it has been the focus of extensive discussion and analysis. In this paper we revisit the NMF optimization problem and analyze its loss landscape in non-worst-case settings. It has recently been observed that gradients in deep networks tend to point towards the final minimizer throughout the optimization. We show that a similar property holds (with high probability) for NMF, provably in a non-worst case model with a planted solution, and empirically across an extensive suite of real-world NMF problems. Our analysis predicts that this property becomes more likely with growing number of parameters, and experiments suggest that a similar trend might also hold for deep neural networks — turning increasing data sets and models into a blessing from an optimization perspective.

## 1 INTRODUCTION

Non-negative matrix factorization (NMF) is a ubiquitous technique for data analysis where one attempts to factorize a measurement matrix $\mathbf{X}$ into the product of non-negative matrices $\mathbf{U}, \mathbf{V}$ (Lee and Seung, 1999). This simple problem has applications in recommender systems (Luo et al., 2014), scientific analysis (Berne et al., 2007; Trindade et al., 2017), computer vision (Gillis, 2012), internet distance prediction (Mao et al., 2006), audio processing (Schmidt et al., 2007) and many more domains. Often, the non-negativity is crucial for interpretability, for example, in the context of crystallography, the light sources, which are represented as matrix factors, have non-negative intensity (Suram et al., 2016).

Like many other non-convex optimization problems, finding the exact solution to NMF is NP-hard (Pardalos and Vavasis, 1991; Vavasis, 2009). NMF's tremendous practical success is however at odds with such worst-case analysis, and simple algorithms based upon gradient descent are known to find good solutions in real-world settings (Lee and Seung, 2001). At the time when NMF was proposed, most analyses of optimization problems within machine learning focused on convex formulations such as SVMs (Cortes and Vapnik, 1995), but owing to the success of neural networks, non-convex optimization has experienced a resurgence in interest. Here, we revisit NMF from a fresh perspective, utilizing recent tools developed in the context of optimization in deep learning. Specifically, our main inspiration is the recent work of Kleinberg et al. (2018) and Zhou et al. (2019) that empirically demonstrate that gradients typically point towards the final minimizer for neural networks trained on real-world datasets and analyze the implications of such convexity properties for efficient optimization.

In this paper, we show theoretically and empirically that a similar property called star-convexity holds in NMF. From a theoretical perspective, we consider an NMF instance with planted solution, inspired by the stochastic block model for social networks (Holland et al., 1983; Decelle et al., 2011) and the planted clique problem studied in sum-of-squares literature (Barak et al., 2016). We prove that between two points the loss is convex with high probability, and conclude that the loss surface is star-convex in the typical case — even if the loss is computed over unobserved data. From an empirical perspective, we verify that our theoretical results hold for an extensive collection

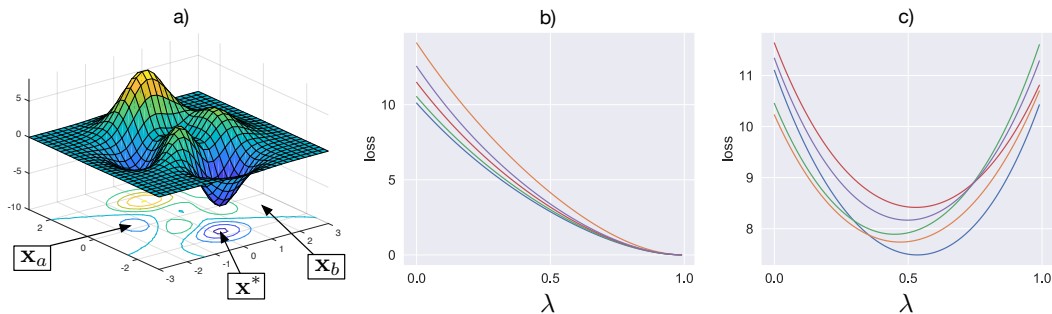

Figure 1: A non-convex loss surface is illustrated in a). In general, the loss will be non-convex on straight paths connecting random points $\mathbf{x}_a, \mathbf{x}_b$ and the global minimizer $\mathbf{x}^*$. We consider a model of NMF with a planted solution, and as shown in b) the loss is typically convex on straight paths between points $\mathbf{x}_a$ and a planted solution $\mathbf{x}^*$. Additionally, as illustrated in c), the loss is typically convex on straight paths between points $\mathbf{x}_a$ and $\mathbf{x}_b$.

of real-world datasets spanning collaborative filtering (Zhou et al., 2008; Kula, 2017; Harper and Konstan, 2016), signal decomposition (Zhu, 2016; Li and Ngom, 2013; Li et al., 2001; Erichson et al., 2018) and audio processing (Flenner and Hunter, 2017; canto Foundation), and demonstrate that the star-convex behavior results in efficient optimization. Finally, we show that star-convex behavior becomes more likely with growing number of parameters, suggesting that a similar result may hold as neural networks become wider. We provide supporting empirical evidence for this hypothesis on modern network architectures.

## 2 NMF AND STAR-CONVEXITY

The aim of NMF is to decompose some large measurement matrix $\mathbf{X} \in \mathcal{R}^{n \times m}$ into two *non-negative* matrices $\mathbf{U} \in \mathcal{R}_+^{n \times r}$ and $\mathbf{V} \in \mathcal{R}_+^{r \times m}$ such that $\mathbf{X} \approx \mathbf{UV}$. The canonical formulation of NMF is

$$\min_{\mathbf{U},\mathbf{V} \geqslant 0} \quad \ell(\mathbf{U}, \mathbf{V}), \text{ where } \ell(\mathbf{U}, \mathbf{V}) = \frac{1}{2}\|\mathbf{UV} - \mathbf{X}\|_F^2 \tag{1}$$

NMF is commonly used in recommender systems where entries $(i, j)$ of $\mathbf{X}$ for example correspond to the rating user $i$ has given to movie $j$ (Luo et al., 2014). In such settings, data might be missing as all users have not rated all movies. In those cases, it is common to only consider the loss over observed data (Zhang et al., 2006; Candès and Recht, 2009). We let $\hat{1}_{(i,j)}$ be an indicator variable that is 1 if entry $(i, j)$ is "observed" and 0 otherwise. The loss function is then

$$\min_{\mathbf{U},\mathbf{V} \geqslant 0} \quad \ell(\mathbf{U}, \mathbf{V}) = \frac{1}{2}\sum_{i,j} \hat{1}_{(i,j)}\left(\left[\mathbf{UV}\right]_{ij} - \mathbf{X}_{ij}\right)^2 \tag{2}$$

NMF is similar to PCA which admits spectral strategies; however, the non-negative constraints in NMF prevent such solutions and result in NP-hardness (Vavasis, 2009). Work on the computational complexity of NMF has shown that the problem is tractable for small constant dimensions $r$ via algebraic methods (Arora et al., 2012). In practice, however, these algorithms are not used, and simple variants of gradient descent, possibly via multiplicative updates (Lee and Seung, 2001), are popular and are known to work reliably (Koren et al., 2009). This gap between theoretical hardness and practical performance is also found in deep learning. Optimizing neural networks is NP-hard in general (Blum and Rivest, 1989), but in practice they can be optimized with simple stochastic gradient descent algorithms to outmatch humans in tasks such as face verification (Lu and Tang, 2015) and playing Atari-games (Mnih et al., 2015). Recent work on understanding the geometry of neural network loss surfaces has promoted the idea of convexity properties. The work of Izmailov et al. (2018) shows that the loss surface is convex around the local optimum, while Zhou et al. (2019) and Kleinberg et al. (2018) show that the gradients during optimization typically point towards the local minima the network eventually converges to. Of central importance in this line of work is **star-convexity**, which is a property of a function $f$ that guarantees that it is convex along straight paths towards the optima $x^*$. See Figure 2 for an example. Formally, it is defined as follows.

**Definition 1.** *A function $f : \mathcal{R}^n \to \mathcal{R}$ is **star-convex** towards $\boldsymbol{x}^*$ if for all $\lambda \in [0, 1]$ and $\boldsymbol{x} \in \mathcal{R}^n$, we have $f\big(\lambda \boldsymbol{x} + (1 - \lambda)\, \boldsymbol{x}^*\big) \leqslant \lambda f(\boldsymbol{x}) + (1 - \lambda) f(\boldsymbol{x}^*)$.*

Optimizing star-convex functions can be done in polynomial time (Lee and Valiant, 2016), in Kleinberg et al. (2018) it is shown how the function only needs to be star-convex under a natural noise model. NMF is not star-convex in general as it is NP-hard, however, it is natural to conjecture that NMF is star-convex in the *typical* case. Such a property could explain the practical success of NMF on real-world datasets, which is not worst-case. This will be the working hypothesis of this paper, where the *typical* case is formalized in Theorem 1. Indeed, one can verify numerically that NMF is typically star-convex for natural distributions and realistically sized matrices, see Figure 1 where we consider a rank 10 decomposition of $(100, 100)$-matrices with iid half-normal entries and a planted solution, sampled as per Assumption 1 given in the next section. Following sections will be dedicated to proving that NMF is star-convex with high probability in a planted model, and to confirm that this phenomenon generalizes to datasets from the real world, which are far from worst-case.

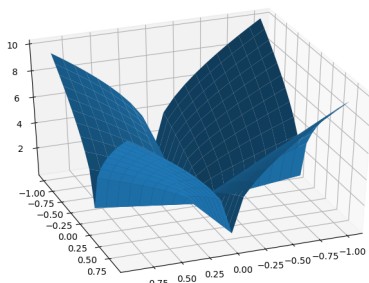

Figure 2: The function $(|x|^p + |y|^p)^{1/p}$ is an example of a star-convex function for $0 < p < 1$. It is non-convex in general, but convex towards $(0, 0)$.

## 3  PROVING TYPICAL-CASE STAR-CONVEXITY

Our aim now is to prove that the NMF loss-function is typically star-convex for natural non-worst-case distribution of NMF instances. We will consider a slighly weaker notation of star-convexity where $f\big(\lambda \mathbf{x} + (1 - \lambda) \mathbf{x}^*\big) \leqslant \lambda f(\mathbf{x}) + (1 - \lambda) f(\mathbf{x}^*)$ holds not for all $\mathbf{x}$, but for random $\mathbf{x}$ with high probability. This is in fact the best achievable, an NMF instance with $u_1 = 1, u^* = 0$ and $v_1 = 0, v^* = 1$ is not star-convex. Our results hold with high probability in high dimensions, similar to Dvoretzky's theorem in convex geometry (Dvoredsky, 1961; Davis).

Inspired by the stochastic block model of social networks (Holland et al., 1983; Decelle et al., 2011) and the planted clique problem (Barak et al., 2016), we will focus on a setting with a planted random solution. In section 4 we verify conclusions drawn from this model transfers to real-world datasets. We will assume that there is a planted optimal solution $(\mathbf{U}^*, \mathbf{V}^*)$, where entries of these matrices are sampled iid from a class of distributions with good concentration properties that include the half-normal distribution and bounded distributions. As is standard in random matrix theory (Vershynin, 2010), we will develop non-asymptotic results that hold with a probability that grows as the matrices of shape $(n, r)$ and $(r, m)$ become large. For this reason, we will need to specify how $r$ and $m$ depend on $n$.

**Assumption 1.** *For $(\mathbf{U}, \mathbf{V}) \in R^{n \times r} \times R^{r \times m}$ we assume that the entries of the matrices $\mathbf{U}, \mathbf{V}$ are sampled iid from a continuous distribution with non-negative support that either $(i)$ is bounded or $(ii)$ can be expressed as a 1-Lipschitz function of a Gaussian distribution. As $n \to \infty$, we assume that $r$ grows as $n^\gamma$ up to a constant factor for $\gamma \in [1/2, 1]$, and $m$ as $n$ up to a constant factor.*

We are now ready to state our main results, that the loss function equation 1 is convex on a straight line between points samples as per Assumption 1, and thus satisfy our slightly weaker notion of star-convexity, with high probability. The probability increases as the size of the problem increases, suggesting a surprising benefit of high dimensionality. We also show similar results for the loss function of equation 2 with unobserved data under the assumption that the event that any entry is observed occurs independently with constant probability $p$. Below we sketch the proof idea and key ingredients, the formal proof is given in Appendix D.

**Theorem 1.** *(**Main**) Let matrices $\mathbf{U}_1, \mathbf{V}_1, \mathbf{U}_2, \mathbf{V}_2, \mathbf{U}^*, \mathbf{V}^*$ be sampled according to Assumption 1. Then there exists positive constants $c_1, c_2$, such that with probability $\geqslant 1 - c_1 \exp(-c_2 n^{1/3})$, the loss function $\ell(\mathbf{U}, \mathbf{V})$ in equation 1 is convex on the straight line $(\mathbf{U}_1, \mathbf{V}_1) \to (\mathbf{U}_2, \mathbf{V}_2)$. The same holds along the line $(\mathbf{U}_1, \mathbf{V}_1) \to (\mathbf{U}^*, \mathbf{V}^*)$. It also holds if any entry $(i, j)$ is observed independently with constant probability $p$, but with probability $\geqslant 1 - c_1 \exp(-c_2 r^{1/3})$.*

### 3.1 PROOF STRATEGY

Let us parametrize the NMF solution one gets along the line $(\mathbf{U}_2, \mathbf{V}_2) \rightarrow (\mathbf{U}_1, \mathbf{V}_1)$ as

$$\hat{\mathbf{X}}(\lambda) = \Big( \lambda \mathbf{U}_1 + (1-\lambda)\mathbf{U}_2 \Big)\Big( \lambda \mathbf{V}_1 + (1-\lambda)\mathbf{V}_2 \Big)$$

Proving Theorem 1 amounts to showing that the loss function $\ell(\lambda) = \frac{1}{2}\|\hat{\mathbf{X}}(\lambda) - \mathbf{X}\|_F^2$ is convex in $\lambda$ with high probability, our strategy is to show that its second-derivate is non-negative. For fixed matrices $\mathbf{U}_1, \mathbf{U}_2, \mathbf{U}^*, \mathbf{V}_1, \mathbf{V}_2, \mathbf{V}^*$, the function $\ell(\lambda)$ is a fourth-degree polynomial in $\lambda$, so its second derivate w.r.t. $\lambda$ will be a second-degree polynomial in $\lambda$. For a general second-degree polynomial $p(x) = ax^2 + bx + c$ we have $p(x) = \frac{1}{a}\big[\big(ax + \frac{b}{2}\big)^2 + \big(ac - \frac{b^2}{4}\big)\big]$. If $a > 0$, which in the case here (see Appendix D), showing that it's positive could be done by showing $ac \geqslant \frac{b^2}{4}$. This is equivalent to showing that

$$2\|\mathbf{W}_2\|_F^2\big(\|\mathbf{W}_1\|_F^2 + 2\langle \mathbf{W}_0, \mathbf{W}_2 \rangle\big) \geqslant 3\big(\langle \mathbf{W}_1, \mathbf{W}_2 \rangle\big)^2 \tag{3}$$

Where the matrices $\mathbf{W}_0, \mathbf{W}_1, \mathbf{W}_2$ are given as $\mathbf{W}_0 = \mathbf{U}_2\mathbf{V}_2 - \mathbf{U}^*\mathbf{V}^*$, $\mathbf{W}_1 = \big(\mathbf{U}_1 - \mathbf{U}_2\big)\mathbf{V}_2 + \mathbf{U}_2\big(\mathbf{V}_1 - \mathbf{V}_2\big)$, $\mathbf{W}_2 = \big(\mathbf{U}_1 - \mathbf{U}_2\big)\big(\mathbf{V}_1 - \mathbf{V}_2\big)$. By slight abuse of notation, we have used $\langle \mathbf{A}, \mathbf{B} \rangle$ to denote $\mathrm{Tr}(\mathbf{AB}^T)$ for matrices $\mathbf{A}, \mathbf{B}$ of the same shape. By exchanging terms in equation 3 by their means one gets

$$2(4rmn\sigma^4)\big(6rmn\sigma^4 + 4rmn\mu_{var}^2\sigma^2 + 2rmn\sigma^4\big) \geqslant 3\big(-4rmn\sigma^4\big)^2 \tag{4}$$

Here, $\sigma^2$ is the variance of the distribution of the entries in the matrices, while $\mu$ is the mean. By just counting terms of order $(rmn\sigma^4)^2$, we see that the LHS has 64 such terms while the RHS has only 48. Thus, if all matrices $\mathbf{W}_0, \mathbf{W}_1$ and $\mathbf{W}_2$ would exactly be equal to their mean, inequality equation 3 would hold. In proving that it holds in general, we will use concentration of measure results from random matrix theory to show that the terms will be concentrated around their means and that large deviations are exponentially unlikely.

**Concentration of measure**  Consider the matrix $\mathbf{W}_2 = \big(\mathbf{U}_1 - \mathbf{U}_2\big)\big(\mathbf{V}_1 - \mathbf{V}_2\big)$. Given that all matrices are iid we can center the variables such that $\mathbf{W}_2 = \big(\mathbf{U}_1 - \mathbf{U}_2\big)\big(\mathbf{V}_1 - \mathbf{V}_2\big) = \big(\bar{\mathbf{U}}_1 - \bar{\mathbf{U}}_2\big)\big(\bar{\mathbf{V}}_1 - \bar{\mathbf{V}}_2\big)$, where the bar denotes the centered matrices. The term $\|\mathbf{W}_2\|_F^2$ can then be written as $\mathrm{Tr}\big(\bar{\mathbf{V}}_1 - \bar{\mathbf{V}}_2\big)^T\big(\bar{\mathbf{U}}_1 - \bar{\mathbf{U}}_2\big)^T\big(\bar{\mathbf{U}}_1 - \bar{\mathbf{U}}_2\big)\big(\bar{\mathbf{V}}_1 - \bar{\mathbf{V}}_2\big)$. Given that all matrix entries are independent as per Assumption 1, we would expect some concentration of measure to hold. Bernstein-type inequalities turns out to be too weak for our purposes, but the field of random matrix theory offer stronger results for matrices with independent sub-Gaussian entries (Ahlswede and Winter, 2002; Tropp, 2012; Guionnet et al., 2000; Meckes and Szarek, 2012). Via such results one can achieve the following concentration result, see Appendix D.

$$P\big(\big|\|\mathbf{W}^2\|_F^2 - \mathbb{E}\big[\|\mathbf{W}_2\|_F^2\big]\big| > t\,r\,n^2\big) \leqslant c_3 \exp\big(-c_4 \min(t^2, t^{1/2})\,n\big) \tag{5}$$

where $c_3, c_4$ are positive constants. In some expression we will however not be able to center all variables, and for such expression one gets similar but slightly weaker concentration results where the exponent scales as $n^{1/3}$ instead of $n$, see Appendix D.

**Proof sketch**  Given that $\mathbb{E}\big[\|\mathbf{W}_2\|_F^2 = 4rmn\sigma^4$, equation 5 says that the probability that the term deviates from its mean by a relative factor $\epsilon$ is less than $c_3 \exp\big(-c_5\epsilon^2 n\big)$ for some small $\epsilon$. By applying similar arguments to terms $\langle \mathbf{W}_0, \mathbf{W}_2 \rangle$ and $\langle \mathbf{W}_1, \mathbf{W}_2 \rangle$, one can show that the probability that they will deviate by a relative factor $\epsilon$ is less than $c_6 \exp(-c_7\epsilon^2 n^{1/3})$. A problematic term is $\|\mathbf{W}_1\|_F^2$ which contains a term of the type $\mathrm{Tr}\big(\bar{\mathbf{V}}_1 - \bar{\mathbf{V}}_2\big)^T \mu_1^T \mu_1 \big(\bar{\mathbf{V}}_1 - \bar{\mathbf{V}}_2\big)$ which has weak concentration properties. Matrices of type $\mathbf{A}^T\mathbf{A}$ are psd, and psd matrices have non-negative trace. Hence, this term will be non-negative, and since it occurs on the LHS of equation 3, we can simply omit it to lower bound the convexity. Using union bound, we bound the probability that at least on term deviate with a relative factor $\epsilon$ by $c_1 \exp(-c_8\epsilon^2 n^{1/3})$ for positive constants $c_1, c_8$. Now, set $\epsilon = 0.01$ If neither variable deviates by a factor of more than $0.01$, then equation 10 still holds since $0.99^2 * 64 \geqslant 1.01^2 * 48$. Thus, inequality is violated with probability at most $c_1 \exp\big(-c_2 n^{1/3}\big)$ for positive $c_1, c_2$. ∎

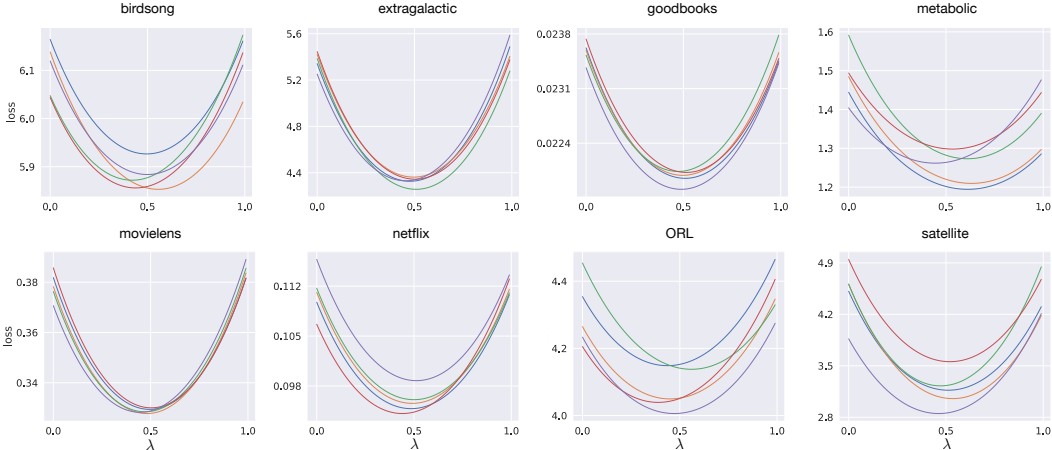

Figure 3: We here illustrate the loss surface of NMF on straight paths connecting two random points for 8 real-world datasets. We overlap 5 independent lines for each dataset. Note that the curves are always convex, suggesting that the loss surface is "typically" convex.

Table 1: Dataset details. References contain suggested rank $r$. See Appendix A for details.

| name | shape $(n, m, r)$ | sparsity | reference |
|---|---|---|---|
| birdsong | $(5120, 1246, 88)$ | | (Flenner and Hunter, 2017) |
| extragalactic | $(2760, 2820, 10)$ | | (Zhu, 2016) |
| goodbooks | $(10000, 43461, 50)$ | 0.0022 | (Kula, 2017) |
| metabolic | $(9335, 36, 3)$ | | (Li and Ngom, 2013) |
| movielens | $(3953, 6041, 20)$ | 0.0419 | (Harper and Konstan, 2016) |
| netflix | $(47928, 8963, 20)$ | 0.0121 | (Zhou et al., 2008) |
| ORL faces | $(400, 10304, 49)$ | | (Li et al., 2001). |
| satellite | $(162, 94249, 4)$ | | (Erichson et al., 2018). |

**Proof sketch for unobserved data**  If the entries in equation 2 are "observed" independently with probability $p$, for fixed matrices $\mathbf{U}_1, \mathbf{U}_2, \mathbf{U}^*, \mathbf{V}_1, \mathbf{V}_2, \mathbf{V}^*$ such that Theorem 1 hold, we have

$$\mathbb{E}[\ell''(\lambda)] = \mathbb{E}\left[ \sum_{ij} \hat{1}_{(i,j)} \left( \hat{\mathbf{X}}'^2_{ij} + \hat{\mathbf{X}}''_{ij}(\hat{\mathbf{X}}_{ij} - \mathbf{X}_{ij}) \right) \right] = p \sum_{ij} \left( \hat{\mathbf{X}}'^2_{ij} + \hat{\mathbf{X}}''_{ij}(\hat{\mathbf{X}}_{ij} - \mathbf{X}_{ij}) \right) \geqslant 0$$

Thus, the expectation of $\ell''(\lambda)$ is convex. To show that it is convex with high probability, one first observes that with high probability, no entry $(i, j)$ in $\ell''(\lambda)$ is particularly large. Assuming this holds via union bound, for fix matrices $\mathbf{U}_1, \mathbf{U}_2, \mathbf{U}^*, \mathbf{V}_1, \mathbf{V}_2, \mathbf{V}^*$ with elements that are "observed" independently with probability $p$, one gets that $\ell''(\lambda)$ is concentrated around its convex mean via Hoeffding bounds (Hoeffding, 1994). ∎

## 4 EXPERIMENTS

### 4.1 VERIFYING THEORETICAL PREDICTIONS

To verify that the conclusions from our planted model hold more broadly, we now consider real-world datasets previously studied in NMF literature. Some have ranks outside the scope of our theoretical model but still display star-convexity properties, indicating that it might be a general phenomenon. We focus on a handful of representative datasets spanning image analysis, scientific applications and collaborative filtering. In Table 1, we list these datasets together with references and sparsity, the decomposition rank we use is based upon values previously used in the literature.

We perform a non-negative matrix factorization via gradient descent, starting with randomly initialized data. To enable comparison between datasets, we scale all data matrices so that the variance of

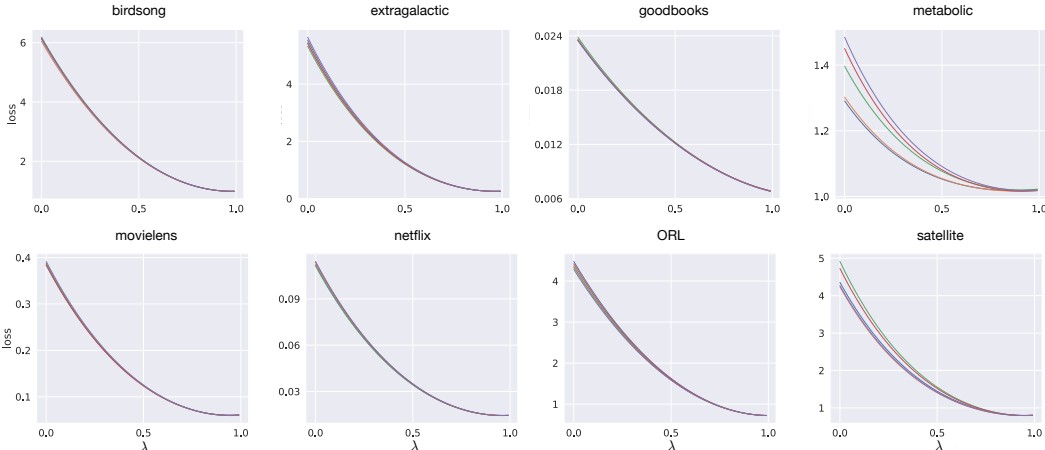

Figure 4: The loss surface of NMF along the straight line from a random point $w_0$ to the local optima $w*$ found via gradient descent from an independent starting point. We overlap 5 independent lines, zoom in for detail. For all datasets, this loss surface is convex, which is in line with what one would expect from Theorem 1.

observed entries is 1 and divide the loss function by the number of (observed) entries. The initialization is performed with half-normal distribution scaled so that the means match with the dataset. For simplicity we use the same gradient descent parameters for all datasets, a learning rate of $1e-7$ with $1e4$ gradient steps which gives convergence for all datasets. For the collaborative filtering datasets (movielens, netflix, and goodbooks) we have unobserved ratings, as is standard in NMF we only compute the loss over observed ratings (Zhang et al., 2006). In Figure 3, we plot the loss function between two random points drawn from the initialization distribution and see that the loss is convex. In Figure 4 we plot the loss function from an initialization point to an independent local optima. These results agree with our planted model; the NMF loss-surface of real-world datasets seem to be largely convex along straight paths. Gradient descent of course only gives local optima but these still display nice star-convexity properties, however, finding the true global optima remains a challenge.

## 4.2 ABLATION EXPERIMENTS

Theorem 1 suggests that as the matrices become larger, NMF is increasingly likely to be star-convex. We are interested to see if this is the case for our real-world datasets, and to this end, we perform ablation experiments varying the dimensions of the matrices. We decrease the number of data points $n$ by subsampling rows and columns uniformly randomly. Our measure of curvature at a point $\mathbf{x}$, given some optimal solution $\mathbf{x}*$, is

$$\alpha(\mathbf{x}) = \min_{\lambda \in [0,1]} \ell''\big(\lambda \mathbf{x}* + (1-\lambda)\mathbf{x}\big) \qquad (6)$$

Note that $\alpha \geqslant 0$ implies star-convexity. In practice, $\mathbf{x}$ and $\mathbf{x}*$ are obtained by random initialization and gradient descent as per the earlier section. For each dataset and subsample rate, we find 100 optima and evaluate the curvature from 50 random points, thus giving 5000 samples of $\alpha$. Figure 5 show how the relative deviation, $\frac{\sigma}{\mu}$, of $\alpha$ decrease as the dataset becomes larger. As shown in Appendix C, the curvature is always positive, thus the curvature becomes increasingly concentrated around its positive mean for larger matrices. We are also interested in investigating whether the loss surface is star-convex during training. In Figure 6, we show that the cosine similarity between the negative gradients $-\nabla\ell(\mathbf{U}, \mathbf{V})$ and the straight line from $\mathbf{x}_0$ to $\mathbf{x}*$ is always positive, and how the loss surface is always star-convex during training. As per (Lee and Valiant, 2016), star-convexity implies efficient optimization. In Figure 7, we illustrate the spectrum of singular values for $\mathbf{U}*$, found for the birdsong dataset, and a random matrix of the same shape with iid entries from a half-normal distribution. The spectra are similar, and while $\mathbf{U}*$ is not random, it seems to share structural qualities with the random matrices used in our proofs. See Appendix C for figures on more datasets.

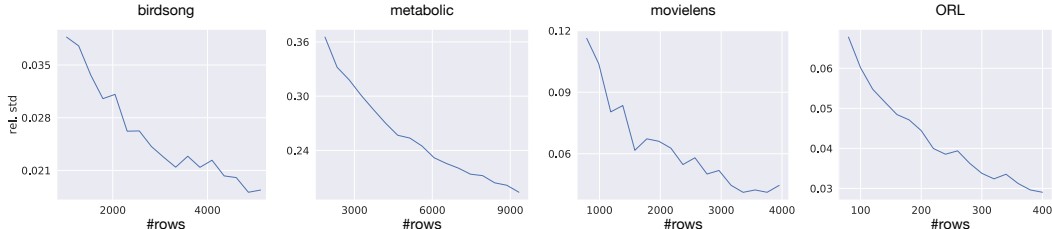

Figure 5: Illustration of how the relative deviation $\sigma/\mu$ of the curvature equation 6 depend on the dataset size. For all datasets, the relative deviations decrease with more samples, suggesting that the (positive) curvature become increasingly concentreated around its mean for larger matrices.

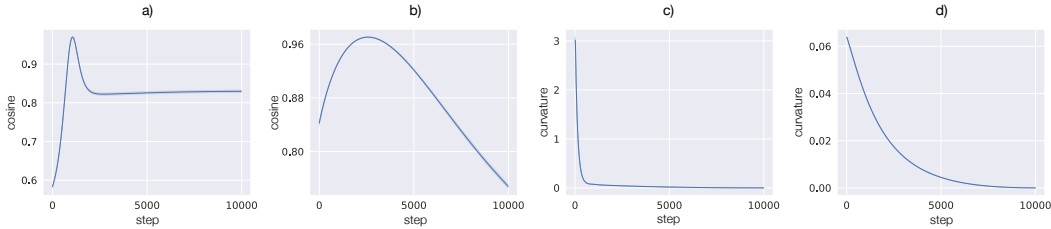

Figure 6: *a*) and *b*) show the cosine similarity between the negative gradient $-\nabla\ell(\mathbf{U}, \mathbf{V})$ and the straight line $(\mathbf{U}_0, \mathbf{V}_0) \rightarrow (\mathbf{U}^*, \mathbf{V}^*)$, during training, for the ORL and movielens dataset. Notice that the cosine similarity is large and non-negative throughout training, showing that the gradients largely follow a straight line. *c*) and *d*) illustrate how the curvature equation 6 varies during training for the ORL and movielens dataset. Note that it is always positive, and thus that the function satisfies star-convexity. Shaded regions represents 95 % confidence computed over 5 iterations.

### 4.3 IMPLICATIONS FOR NEURAL NETWORKS

We have seen how increasing the number of parameters makes NMF problems more likely to be star-convex, and Figure 5 shows how more parameters make the curvature tend towards its positive mean. Theorem 1 suggest that this is a result of concentration of measure, and it is natural to believe that similar phenomenon would persist in the context of neural networks. It has previously been observed how neural networks are locally convex (Izmailov et al., 2018; Kleinberg et al., 2018), and also how overparameterization is important in deep learning (Arora et al., 2018; Frankle and Carbin, 2018). Based upon our observations in NMF, we hypothesise that a major benefit of overparameterization is in making the loss surface more convex via concentration of measure w.r.t. the weights.

To verify this hypothesis, we consider image classification on CIFAR10 (Krizhevsky et al., 2014) with Resnet networks (He et al., 2016) trained with standard parameters (see Appendix B). Networks are typically only locally convex, a property we quantify as the length of subsets of random "lines" in parameter space, along which the loss is convex. We sample a random direction $\mathbf{r}$ and then consider a subspace of length one along this direction centered around the current parameters $\mathbf{w}$. We then define the convexity length scale as the length of the maximal sub-interval $[\lambda_1, \lambda_2]$ containing 0 on which $\ell(\mathbf{w} + \lambda\mathbf{r})$ is convex. Directions are sampled from Gaussian distributions and then normalized for each filter $f$ to have the same norm as the weights of $f$. Table 2 show how this length-scale of local convexity varies with depth, width, and training, where width is varied by multiplying the number of channels by $k$. Increased width indeed makes the landscape increasingly locally convex for all but the most shallow networks, results that support our hypothesis. See Appendix B for extended tables.

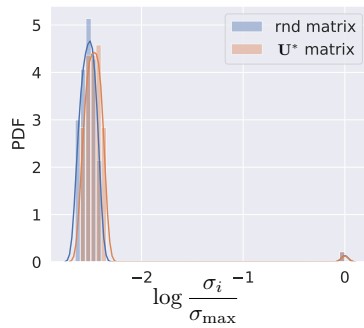

Figure 7: Histogram of singular values $\sigma_i$ for found $\mathbf{U}^*$ and a random matrix. Note the similarity of the spectrum. Best viewed in color.

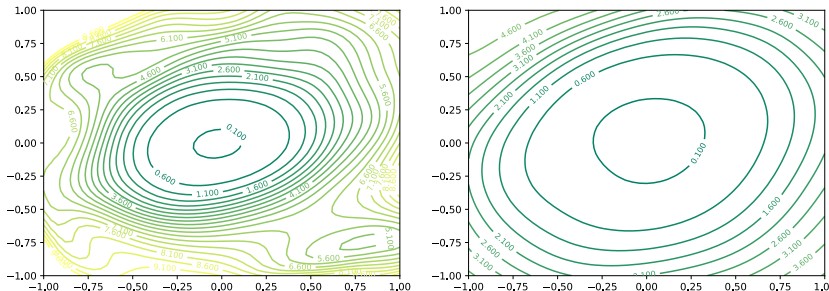

Figure 8: The loss landscape of a 110-layer Resnet architecture at epoch 200 along two random directions, visualized as in Li et al. (2018). The network in the right image is four times as wide, and its loss landscape is increasingly convex. In Table 2 we generalize this idea, showing that the length scale of local convexity increases with network width.

Table 2: Typical length scales of local convexity for Resnet networks with various depth and width (indicated by $k$). We sample 25 random "lines" in parameter space of length 1 centered on current parameters, and report average length of convex subset of such "lines". Increased width makes the loss surface increasingly locally convex, while convexity decreases with depth and during training.

| epoch | 20-layers | | | 44-layers | | | 68-layers | | | 110-layers | | |
|---|---|---|---|---|---|---|---|---|---|---|---|---|
| | k=1 | k=2 | k=4 | k=1 | k=2 | k=4 | k=1 | k=2 | k=4 | k=1 | k=2 | k=4 |
| 0 | 0.96 | **1.0** | **1.0** | **1.0** | **1.0** | **1.0** | **1.0** | **1.0** | **1.0** | 0.94 | 0.94 | 0.94 |
| 100 | **0.87** | 0.84 | **0.87** | 0.72 | 0.79 | **0.83** | 0.71 | 0.76 | **0.87** | 0.79 | 0.75 | **0.91** |
| 200 | **0.81** | 0.74 | 0.73 | 0.66 | 0.68 | **0.76** | 0.6 | 0.71 | **0.8** | 0.71 | 0.71 | **0.82** |
| 300 | 0.71 | 0.72 | **0.75** | 0.57 | 0.68 | **0.78** | 0.58 | 0.67 | **0.81** | 0.63 | 0.68 | **0.82** |

## 5 RELATED WORK

Our work was initially motivated by findings in Kleinberg et al. (2018) and Zhou et al. (2019) regarding star-convexity in neural networks. As the success of deep learning has become apparent, researchers have empirically investigated neural network trained on real-world datasets (Li and Yuan, 2017; Keskar et al., 2016). In the context of sharp vs flat local minima (Keskar et al., 2016), Li et al. (2018) illustrate how width improved flatness in a Resnet network, an observation Table 2 quantifies. In general, real-world datasets seem to be far more well-behaved than the worst case, given that training neural networks are NP-hard (Blum and Rivest, 1989). There is extensive work on non-worst-case analysis of algorithms and machine learning models, and on what problem distributions can guarantee tractability which addresses such gaps (Bilu and Linial, 2012; Afshani et al., 2017). On the positive side Arora et al. (2012) has shown an exact algorithm for NMF that runs in polynomial time for small constant $r$, and there are positive results for 'separable' NMF (Donoho and Stodden, 2004). Compressive sensing (Candes et al., 2004), smoothed analysis (Spielman and Teng, 2004) and problems with "planted" solutions (Barak et al., 2016) (Holland et al., 1983) similarly makes assumptions on the input. Researchers have also been interested in theoretical convergence properties of shallow and linear networks (Lu and Kawaguchi, 2017), where a common theme is that functions with only saddle points and global minima can be effectively optimized (Ge et al., 2015). In analysis of neural networks, random matrix theory often plays a role, directly or indirectly (Choromanska et al., 2015; Glorot and Bengio, 2010; Pennington and Bahri, 2017; Xiao et al., 2018).

## 6 CONCLUSIONS

This paper revisits NMF, a non-convex optimization problem in machine learning. We have shown that NMF is typically star-convex, provably for a natural average-case model and empirically on an extensive set of real-world datasets. Our results support the counter-intuitive observation that optimization might sometimes be *easier* in higher dimensions due to concentration of measure effects.

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

## A  DATASETS

Here we describe the datasets we used to evaluate our results on and provide references for them.

**Birdsong**: time series for bird calls, see (Flenner and Hunter, 2017; canto Foundation).
**Extragalactic**: dataset for extragalactic light sources, see (Zhu, 2016).
**Goodbooks**: user ratings for books, see (Kula, 2017; Bourgin, 2018).
**Metabolic**: Yest metabolic activity time series, see (Li and Ngom, 2013).
**Movielens**: user ratings for movies , see (Harper and Konstan, 2016; Li et al., 2012; Zhang et al., 2006).
**Netflix**: User ratings for movies, see (Zhou et al., 2008; Ampazis, 2008; Li et al., 2012).
**ORL faces**: black and white facial images, see (Li et al., 2001; Hoyer, 2004).
**Satellite**: satellite urban spectral-image, see (Erichson et al., 2018; Gillis, 2014).

## B  DNN PARAMETERS

Table 3 presents the hyper-parameters used for training the neural networks. Data augmentation consists of randomized cropping with 4-padding and randomized horizontal flipping. The learning rate is decreased by a factor of 10 after epochs $(150, 225, 275)$. We use the standard Resnet architecture for CIFAR10 with three segments with $(16, 32, 64)$ channels each, and use the standard blocks (i.e. not bottleneck blocks) (He et al., 2016).

Table 3: Hyper-parameters used for training.

| Parameter | Value |
|---|---|
| init. learning rate | 0.1 |
| SGD momentum | 0.9 |
| batch size | 128 |
| weight decay | 0.0005 |
| initialization | Kaiming |
| loss | cross-entropy |

## C  EXTENDED NMF FIGURES

## D  PROOFS

### D.1  OVERVIEW

We here provide the proof of Theorem 1. In section D.2 we present the notation and derive the main inequality. In Section D.3, we prove that the loss function is convex between any two random points with high probability. Section D.4, we prove that this also holds towards the planted solution. For the case with unobserved data, we prove that the loss function is convex between any two random points with high probability in Section D.5. That this holds towards the planted solution in the case of unobserved data follows from previous sections, and the proof of this fact omitted. Together, these results form Theorem 1 as stated in the main text. Section D.6 presents the concentration

Table 4: Continuation of Table 2.

|  | 32-layers | | | 56-layers | | | 80-layers | | |
|---|---|---|---|---|---|---|---|---|---|
| epoch | k=1 | k=2 | k=4 | k=1 | k=2 | k=4 | k=1 | k=2 | k=4 |
| 0 | **1.0** | **1.0** | **1.0** | **1.0** | **1.0** | **1.0** | **1.0** | **1.0** | **1.0** |
| 100 | 0.77 | 0.8 | **0.84** | 0.7 | 0.81 | **0.84** | 0.72 | **0.85** | 0.81 |
| 200 | 0.61 | 0.68 | **0.8** | 0.63 | 0.67 | **0.8** | 0.59 | 0.7 | **0.77** |
| 300 | 0.55 | 0.68 | **0.82** | 0.57 | 0.66 | **0.79** | 0.63 | 0.71 | **0.79** |

Table 5: Table 2 for architectures without skip-connections

| epoch | 20-layers | | | 32 | | | 44-layers | | |
|---|---|---|---|---|---|---|---|---|---|
| | k=1 | k=2 | k=4 | k=1 | k=2 | k=4 | k=1 | k=2 | k=4 |
| 0 | 0.15 | **0.62** | 0.15 | 0.2 | 0.84 | **1.0** | 0.43 | 0.33 | **1.0** |
| 100 | 0.71 | 0.79 | **0.93** | 0.68 | 0.7 | **0.83** | 0.57 | 0.61 | **0.6** |
| 200 | 0.74 | 0.67 | **0.8** | 0.63 | 0.63 | **0.86** | 0.57 | 0.6 | **0.67** |
| 300 | 0.69 | 0.63 | **0.82** | 0.5 | 0.63 | **0.84** | 0.47 | 0.56 | **0.67** |

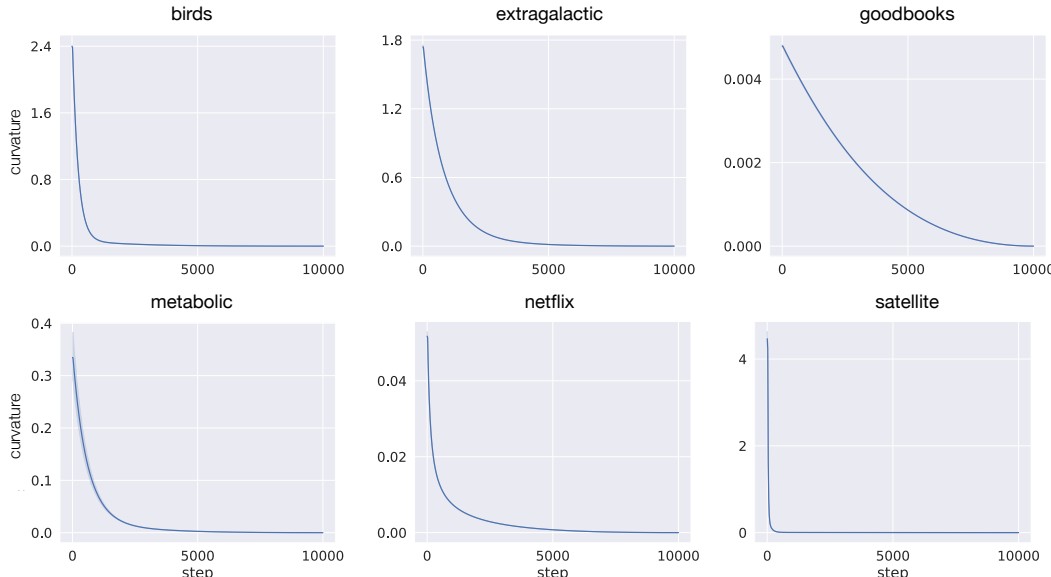

Figure 9: The curvature equation 6 during training, as in Figure 6, for various datasets. Note that it is always positive, and thus that the function satisfies star-convexity. Note the small shaded regions, which represents the 95 % confidence computed over 5 repetitions.

Table 6: Links for used datasets.

| dataset | link |
|---|---|
| birdsong | https://www.kaggle.com/rtatman/british-birdsong-dataset/data |
| extragalactic | https://s3.us-east-2.amazonaws.com/setcoverproblem/Extra... |
| | ...galacticTest/Extragalatic_Archetype_testsample_spec.fits |
| goodbooks | https://www.kaggle.com/zygmunt/goodbooks-10k |
| metabolic | ftp://ftp.ncbi.nlm.nih.gov/geo/series/GSE3nnn |
| | /GSE3431/matrix/GSE3431_series_matrix.txt.gz |
| movielens | https://www.kaggle.com/prajitdatta/movielens-100k-dataset |
| netflix | https://www.kaggle.com/netflix-inc/netflix-prize-data |
| ORL | https://github.com/chibuta |
| | /2Layer_convolution/blob/master/ORL_faces.npz.zip |
| satellite | http://www.escience.cn/system/file?fileId=69117 |

results we need. The proofs include considerable amounts of elementary algebraic calculations. For completeness and ease of reference, we present these in Appendix E.

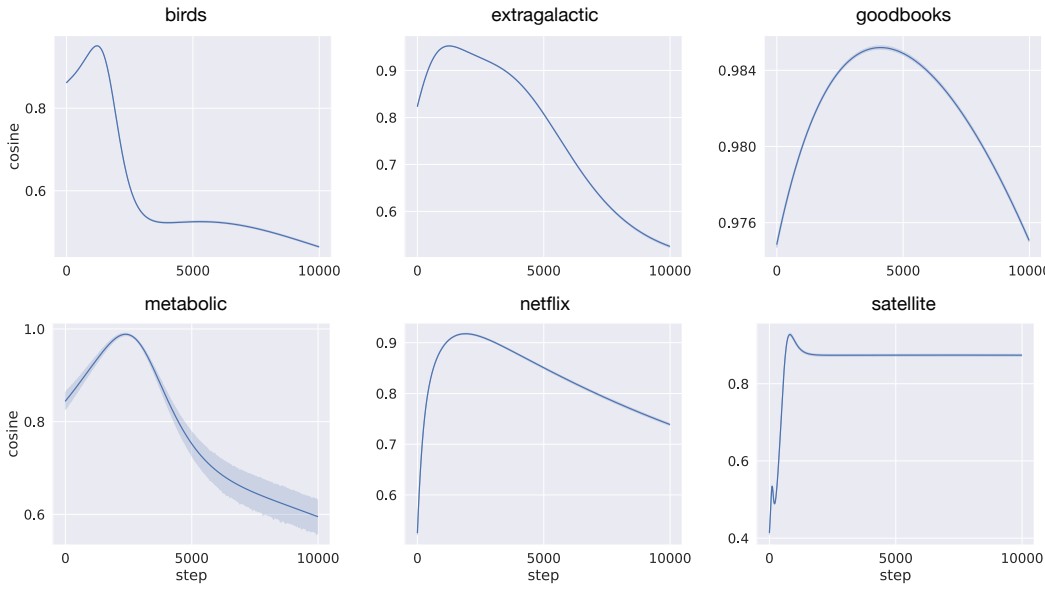

Figure 10: The cosine similarity between the negative gradient $-\nabla\ell(\mathbf{U}, \mathbf{V})$ and the straight line $(\mathbf{U}_0, \mathbf{V}_0) \rightarrow (\mathbf{U}^*, \mathbf{V}^*)$, during training as in Figure 6. Note the small shaded regions, which represents the 95 % confidence computed over 5 repetitions.

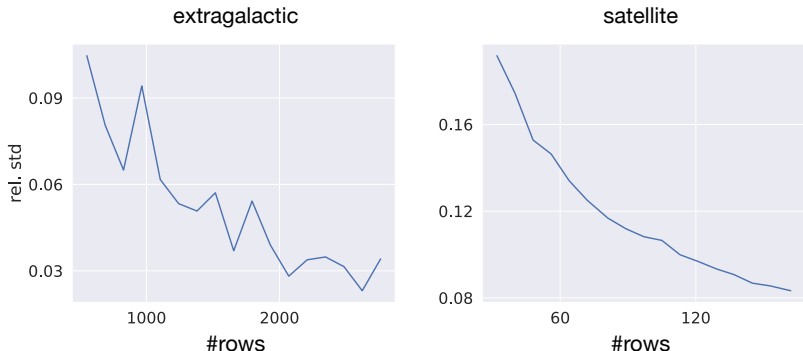

Figure 11: Illustration of how the relative deviation $\sigma/\mu$ of the curvature equation 6 depend on the dataset size as in Figure 6. Unfortunately, computing these figures for the larger Netflix and Goodbooks datasets is computationally infeasible for us.

### D.2 NOTATION AND THE MAIN INEQUALITY

The proof will involve various constants $c_0, c_1, c_2$ that do not depend on $n$. For convenience, we will let exact values be context dependent and will reuse the symbols $c_0, c_1, c_2$ and so on. The proofs will not depend on the exact values of these constants. Constants in our main result can be improved slightly, but at the cost of a more complicated expression. We present the non-optimized version. We will let boldface capital letters denote matrices, boldface lower-case letter vectors and non-boldface letters denote scalars. Let $\mathbf{U}_1$, $\mathbf{U}_2$, $\mathbf{U}^*$ be $n$-by-$r$ matrices and $\mathbf{V}_1$, $\mathbf{V}_2$, $\mathbf{V}^*$ $r$-by-$m$ matrices sampled as per Assumption 1. Without loss of generality, we assume $m = nc_m$ where $0 < c_m \leqslant 1$. We will let $\mu_{var}$ be the expectation, and $\sigma^2$ be the variance of the entry-wise distributions. We will denote the centered variables by $\bar{\mathbf{U}}_1, \bar{\mathbf{V}}_1$ and so on, and the mean matrix of $\mathbf{U}$ and $\mathbf{V}$ resp as $\mathbf{1}_{n \times r}, \mathbf{1}_{r \times m}$. Then $\mathbf{U} = \bar{\mathbf{U}} + \mathbf{1}_{n \times r}$. By slight abuse of notation, we will use the convention $\langle \mathbf{X}, \mathbf{Y} \rangle = \text{Tr}\left(\mathbf{X}\mathbf{Y}^T\right)$ for matrices $\mathbf{X}, \mathbf{Y}$ of the same shape. The loss function we wish to minimize, subject to non-negativity constraints, is

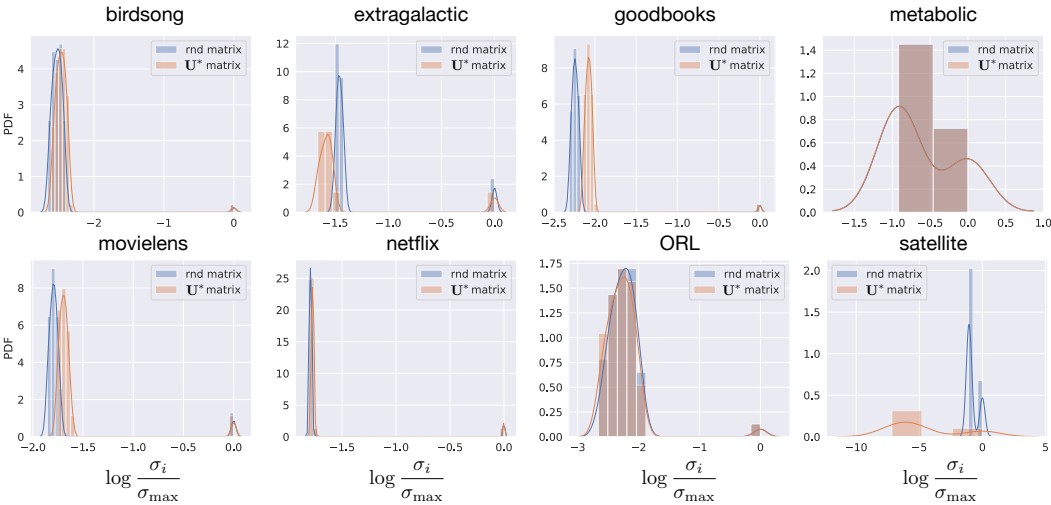

Figure 12: Histogram of singular value spectra for **U\*** found via gradient descent and a random matrix of the same shape with entries drawn iid from a half-normal distribution. For the datasets with larger ranks, the spectra typically agree significantly.

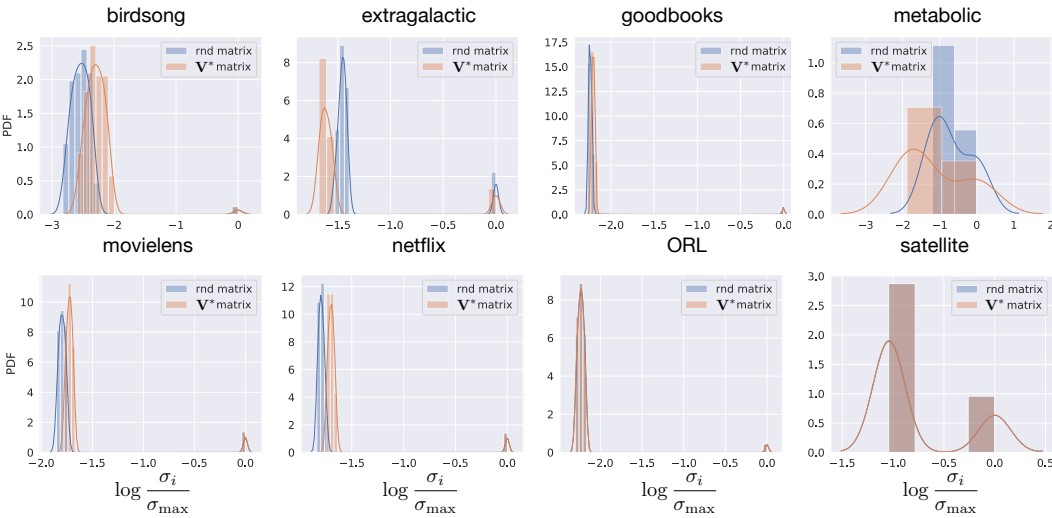

Figure 13: Histogram of singular value spectra for **V\*** found via gradient descent and a random matrix of the same shape with entries drawn iid from a half-normal distribution. For the datasets with larger ranks, the spectra typically agree significantly.

$$\ell(\mathbf{U}, \mathbf{V}) = \frac{1}{2}\|\mathbf{U}\mathbf{V} - \mathbf{X}\|_F^2 \tag{7}$$

When we interpolate the solution $\mathbf{U}\mathbf{V}$ between any fixed two points $(\mathbf{U}_1, \mathbf{V}_1)$ and $(\mathbf{U}_2, \mathbf{V}_2)$, we write

$$\hat{\mathbf{X}}(\lambda) = \big(\lambda\mathbf{U}_1 + (1-\lambda)\mathbf{U}_2\big)\big(\lambda\mathbf{V}_1 + (1-\lambda)\mathbf{V}_2\big)$$

The loss is then

$$\ell(\lambda) = \frac{1}{2}\|\hat{\mathbf{X}}(\lambda) - \mathbf{X}\|_F^2$$

Proving Theorem 1 in the case of two random points and completely observed data amounts to showing that the continuous function $\ell(\lambda)$ is convex in $\lambda$ on the interval $[0, 1]$ with high probability. We will show that the second-derivate is non-negative. For fixed matrices $\mathbf{U}_1, \mathbf{U}_2, \mathbf{U}^*, \mathbf{V}_1, \mathbf{V}_2, \mathbf{V}^*$, the loss $\ell(\lambda)$ is a fourth-degree polynomial in $\lambda$, so its second derivate w.r.t. $\lambda$ will be a second-degree polynomial. For a general second-degree polynomial $p(x) = ax^2 + bx + c$ we have $p(x) = \frac{1}{a}\big[\big(ax + \frac{b}{2}\big)^2 + \big(ac - \frac{b^2}{4}\big)\big]$. By inspecting equation 8 one can conclude that $a > 0$. Showing that the polynomial is non-negative can then be done by showing $ac \geqslant \frac{b^2}{4}$. Let $\hat{\mathbf{X}}_{ij}(\lambda)$ be element $i, j$ of the matrix $\hat{X}(\lambda)$. The derivates of $\ell(\lambda)$ are then

$$\ell(\lambda) = \frac{1}{2}\sum_{ij}\big(\hat{\mathbf{X}}_{ij}(\lambda) - \mathbf{X}_{ij}\big)^2$$

$$\ell'(\lambda) = \sum_{ij}\big(\hat{\mathbf{X}}_{ij}(\lambda) - \mathbf{X}_{ij}\big)\hat{\mathbf{X}}'_{ij}(\lambda)$$

$$\ell''(\lambda) = \sum_{ij}\hat{\mathbf{X}}''_{ij}(\lambda)\big(\hat{\mathbf{X}}_{ij}(\lambda) - \mathbf{X}_{ij}\big) + \hat{\mathbf{X}}'_{ij}(\lambda)^2$$

$$\ell'''(\lambda) = 3\sum_{ij}\hat{\mathbf{X}}''_{ij}(\lambda)\hat{\mathbf{X}}'_{ij}(\lambda)$$

$$\ell''''(\lambda) = 3\sum_{ij}\hat{\mathbf{X}}''_{ij}(\lambda)^2 \tag{8}$$

Where we have

$$\hat{\mathbf{X}}(\lambda) = \big(\lambda\mathbf{U}_1 + (1-\lambda)\mathbf{U}_2\big)\big(\lambda\mathbf{V}_1 + (1-\lambda)\mathbf{V}_2\big)$$

$$\hat{\mathbf{X}}'(\lambda) = \Big(\mathbf{U}_1 - \mathbf{U}_2\Big)\Big(\lambda\mathbf{V}_1 + (1-\lambda)\mathbf{V}_2\Big) + \Big(\lambda\mathbf{U}_1 + (1-\lambda)\mathbf{U}_2\Big)\Big(\mathbf{V}_1 - \mathbf{V}_2\Big)$$

$$\hat{\mathbf{X}}''(\lambda) = 2\Big(\mathbf{U}_1 - \mathbf{U}_2\Big)\Big(\mathbf{V}_1 - \mathbf{V}_2\Big)$$

Let us define

$$\mathbf{W}_0 = \mathbf{U}_2\mathbf{V}_2 - \mathbf{U}^*\mathbf{V}^*$$

$$\mathbf{W}_1 = \Big(\mathbf{U}_1 - \mathbf{U}_2\Big)\mathbf{V}_2 + \mathbf{U}_2\Big(\mathbf{V}_1 - \mathbf{V}_2\Big)$$

$$\mathbf{W}_2 = \Big(\mathbf{U}_1 - \mathbf{U}_2\Big)\Big(\mathbf{V}_1 - \mathbf{V}_2\Big)$$

By expanding $\ell''(\lambda)$ in a McLaurin series around 0, we see that proving non-negativity of $\ell''(\lambda)$ can be done by proving $2\ell(0)''\ell''''(0) \geqslant \ell'''(0)^2$. Thus, proving Theorem 3 then amounts to showing that

$$2\|\mathbf{W}_2\|_F^2\big(\|\mathbf{W}_1\|_F^2 + 2\langle\mathbf{W}_0, \mathbf{W}_2\rangle\big) \geqslant 3\big(\langle\mathbf{W}_1, \mathbf{W}_2\rangle\big)^2 \tag{9}$$

### D.3 PROVING THE INEQUALITY BETWEEN RANDOM POINTS

**Theorem 2.** *Let $(\mathbf{U}_1, \mathbf{V}_1)$, $(\mathbf{U}_2, \mathbf{V}_2)$ and the planted solution $(\mathbf{U}^*, \mathbf{V}^*)$ be sampled as per Assumption 1. The loss function equation 7 is convex on a straight line connecting $(\mathbf{U}_1, \mathbf{V}_1)$ and $(\mathbf{U}_2, \mathbf{V}_2)$ with probability $\geqslant 1 - c_1\exp\big(-c_2 n^{1/3}\big)$ for positive constants $c_1, c_2$.*

**Proof.** As per section D.2, we only need to prove that equation 9 holds with probability $\geqslant 1 - c_1\exp\big(-c_2 n^{1/3}\big)$. Let us exhange all terms in equation 9 with their means, which are given in Facts 7, 5, 6 and 8

$$2(4rmn\sigma^4)\big(6rmn\sigma^4 + 4rmn\mu_{var}^2\sigma^2 + 2rmn\sigma^4\big) \geqslant 3\big(-4rmn\sigma^4\big)^2 \tag{10}$$

Clearly, equation 9 would hold if one naively exchanges the terms for their means. Just counting terms of order $rmn\sigma^4$ we have 64 on the LHS and 48 on the RHS. We need to show that deviations sufficiently large to violate the inequality are unlikely. Our strategy will be to show that the factors are exponentially unlikely to deviate substantially individually, and then use union bounds. The concentration results and their derivations are detailed in Section D.6.

Consider the random variable $\|\mathbf{W}_2\|_F^2$. As per equation 14, $\|\mathbf{W}_2\|_F^2$ can be rewritten into polynomials of centered random matrices where we have to pad the matrices with rows/columns of all zeros. We can then apply the concentration result of Fact 1 to conclude that the probability that $\|\mathbf{W}_2\|_F^2$ deviates from its mean by a factor $(1 + \epsilon)$ is no more than $c_1\exp(-c_2\epsilon^2 n)$.

Now, consider the random variable $\|\mathbf{W}_1\|_F^2$. As in the proof of Fact 7, $\|\mathbf{W}_1\|_F^2$ can be rewritten as the sum of expression equation 16 plus terms of centered variables. The latter expressions are polynomials in centered random matrices, after padding some columns/rows with zeros we can apply Fact 1 to conclude that the probability that it deviates from its mean by a factor $(1 + \epsilon)$ is no more than $c_1\exp(-c_2\epsilon^2 n)$. Consider the expression equation 16. Clearly the expressions $\mathbf{1}_{r\times m}^T\big(\bar{\mathbf{U}}_1 - \bar{\mathbf{U}}_2\big)^T\big(\bar{\mathbf{U}}_1 - \bar{\mathbf{U}}_2\big)\mathbf{1}_{r\times m}$ and $\big(\bar{\mathbf{V}}_1 - \bar{\mathbf{V}}_2\big)^T\mathbf{1}_{n\times r}^T\mathbf{1}_{n\times r}\big(\bar{\mathbf{V}}_1 - \bar{\mathbf{V}}_2\big)$ will be psd, and thus their trace will be nonnegative. We can thus lower bound the LHS of equation 10 by omitting the terms in equation 16, and will thus only consider terms scaling as $rn^2\sigma^4$.

Now, consider the random variable $\langle\mathbf{W}_0, \mathbf{W}_2\rangle$. As per equation 19, the variable $\langle\mathbf{W}_0, \mathbf{W}_2\rangle$ can also be separated into two parts. After padding with some zero rows/columns to obtain square matrices, the first is a polynomial in centered matrices for which Fact 1 applies which again bounds the probability of deviations by a factor $(1 + \epsilon)$ by $c_1\exp(-c_2\epsilon^2 n)$. The second one is a sum of the type $\mathbf{1}\mathbf{X}_1\mathbf{X}_2\mathbf{X}_3$ modulo permutations, under which the trace is invariant. We can thus apply Fact 2 which bounds the probability for deviations of size $\epsilon\langle\mathbf{W}_0, \mathbf{W}_2\rangle$ by $c_1\exp\big(-\epsilon c_2 n^{1/3}\big)$.

At last, consider the random variable $\langle\mathbf{W}_1, \mathbf{W}_2\rangle$. By equation 18 can, just like $\langle\mathbf{W}_0, \mathbf{W}_2\rangle$, be written as the sum of a polynomial in centered random matrices, and sums of variables of the type $\mathbf{1}\mathbf{X}_1\mathbf{X}_2\mathbf{X}_3$ modulo permutations. We can thus apply the same argument as for $\langle W_0, W_2\rangle$ to bounds the probability for deviations of relative size $\epsilon$ by $c_1\exp\big(-\epsilon c_2 n^{1/3}\big)$.

We can now apply union bound on the variables $\|\mathbf{W}_1\|_F^2$, $\|\mathbf{W}_2\|_F^2$, $\langle\mathbf{W}_0, \mathbf{W}_2\rangle$ and $\langle\mathbf{W}_1, \mathbf{W}_2\rangle$, modulu term equation 16, deviating from their respective mean by a factor $\epsilon$. The probability that either of them does is no more than $c_1\exp\big(-c_2\epsilon^2 n^{1/3}\big)$ by union bounds. Now, set $\epsilon = 0.01$ If neither variable deviates by a factor of more than 0.01, then equation 10 still holds if we only count terms scaling as $rmr\sigma^4$ since $0.99^2 * 64 \geqslant 1.01^2 * 48$. Thus, inequality is violated with probability at most $c_1\exp\big(-c_2 n^{1/3}\big)$ for positive constants $c_1, c_2$. ∎

### D.4 PROVING THE INEQUALITY TOWARDS THE PLANTED SOLUTION

**Theorem 3.** *Let $(\mathbf{U}_1, \mathbf{V}_1)$ and the planted solution $(\mathbf{U}^*, \mathbf{V}^*)$ be sampled as per Assumption 1. The loss function of equation 7 is convex on a straight line connecting $(\mathbf{U}_1, \mathbf{V}_1)$ and $(\mathbf{U}^*, \mathbf{V}^*)$ with probability $\geqslant 1 - c_1 \exp\left(-c_2 n^{1/3}\right)$ for constants $c_1, c_2 > 0$.*

**Proof.** To prove this, we can repeat the argument used in the proof of Theorem 2. The terms $\|\mathbf{W}_2\|^2$, $\|\mathbf{W}_1\|^2$ and $\langle \mathbf{W}_1, \mathbf{W}_2 \rangle$ will have the same mean and concentration since $\mathbf{U}_1$ and $\mathbf{U}^*$ are identically distributed. The only difference will be the term $\langle \mathbf{W}_0, \mathbf{W}_2 \rangle$, which now will have mean $2rmn\sigma^4$ per Fact 10. This will only make the LHS of equation 10 larger, so the inequality will still hold and the concentration of measure properties will ensure that equation 9 holds with high probability as in the proof of theorem 2. ∎

### D.5 PROVING THE INEQUALITY BETWEEN RANDOM POINTS FOR UNOBSERVED DATA

As explained in the main text, we will assume that entries are observed with independent probability $p$. We formalize this in the following assumption

**Assumption 2.** *For any $n$, let $\hat{1} \in \{0, 1\}^{n \times m}$ have entries iid drawn from a Bernoulli distribution with probability $p$, which is constant w.r.t $n$.*

Given any set of observations $\hat{1}$, we define our loss function as

$$\ell(\mathbf{U}, \mathbf{V}) = \sum_{i,j} \hat{1}_{(i,j)} \left( \left[ \mathbf{U}\mathbf{V} \right]_{ij} - \mathbf{X}_{ij} \right)^2 \tag{11}$$

**Theorem 4.** *Let $(\mathbf{U}_1, \mathbf{V}_1)$, $(\mathbf{U}_2, \mathbf{V}_2)$ and the planted solution $(\mathbf{U}^*, \mathbf{V}^*)$ be sampled as per Assumption 1. Let the set of observations $\hat{1}$ be sampled as per Assumption 2. The loss function equation 11 is convex on a straight line connecting $(\mathbf{U}_1, \mathbf{V}_1)$ and $(\mathbf{U}_2, \mathbf{V}_2)$ with probability $\geqslant 1 - c_1 \exp\left(-c_2 r^{1/3}\right)$ for constants $c_1, c_2 > 0$.*

**Proof.** If no entries are observed, we can claim trivially that the function is convex. If some entries are observed, then $a > 0$ since $\hat{\mathbf{X}}''_{ij}(\lambda)^2 > 0$ for any $(i, j)$ with probability 1. The second derivate of equation 11 can be written as

$$\ell''(\lambda) = \sum_{ij} \hat{1}_{(i,j)} \left( \hat{\mathbf{X}}'^2_{ij} + \hat{\mathbf{X}}''_{ij} (\hat{\mathbf{X}}_{ij} - \mathbf{X}_{ij}) \right)$$

Using Lemma 1, we can assume by union bound that no entry in $\hat{\mathbf{X}}, \hat{\mathbf{X}}', \hat{\mathbf{X}}'', \mathbf{X}$ is larger than $\mathcal{O}(r^{2/3})$ with probability $\leqslant c_1 n^2 \exp\left(-c_2 r^{1/3}\right)$. Note that this estimate gives the $\exp\left(-c_2 r^{1/3}\right)$ scaling of the proof. Let assume that the entries are bounded like this, then no entry in $\hat{\mathbf{X}}'^2_{ij} + \hat{\mathbf{X}}''_{ij} (\hat{\mathbf{X}}_{ij} - \mathbf{X}_{ij})$ will have magnitude more than $\mathcal{O}(r^{4/3})$ for any $\lambda$. Standard Hoeffding bounds (Hoeffding, 1994) states that for variables $\{x_i\}$ bounded in $[A_i, B_i]$ we have

$$P\left( \left| \frac{1}{n} \sum_i^n x_i \right| \geqslant t \right) \leqslant \exp\left( -\frac{2n^2 t^2}{\sum_i^n (b_i - a_i)^2} \right)$$

Let us define

$$y(\lambda) = \sum_{ij} \hat{\mathbf{X}}'^2_{ij}(\lambda) + \hat{\mathbf{X}}''_{ij} (\hat{\mathbf{X}}_{ij}(\lambda) - \mathbf{X}_{ij})$$

Let us now consider fixed matrices $\mathbf{U}_1, \mathbf{U}_2, \mathbf{U}^*, \mathbf{V}_1, \mathbf{V}_2, \mathbf{V}^*$ satisfying Lemma 1, but let $\hat{1}$ remain a random variable. We can then consider the product of any two variables out of $\hat{\mathbf{X}}_{ij}, \hat{\mathbf{X}}'_{ij}, \hat{\mathbf{X}}''_{ij}, \mathbf{X}_{ij}$ as fixed constants in the interval $[-c_1 r^{4/3}, c_1 r^{4/3}]$ for some constant $c_1$. We note that $y(\lambda)$ is the

sum of $c_m n^2$ variables, each of size at most $\mathcal{O}(r^{4/3})$. Now $y(\lambda)$ is a sum of independent variables and Hoeffdinger applies with $(b_i - a_i)^2 \leqslant \mathcal{O}(r^{8/3})$ for all $i$. Taking $t = cr^{2/3}$ gives us

$$P\left(\left|y(\lambda) - \mathbb{E}[y(\lambda)]\right| \geqslant cr^{2/3}n^2\right) \leqslant \exp\left(-\frac{2c^2 c_m n^4 r^{4/3}}{n^2 r^{8/3}}\right) \leqslant \exp\left(-cn^{2/3}\right)$$

This will hold for any $\lambda$. We can assume that this holds for let's say 3 evenly spaced $\lambda$ by union bound. Using union bounds, we assume the following: 1) $y(\lambda) > c_1 rmn$ for all $\lambda$, which follows from the derivation of Theorem 2 with probability $\geqslant 1 - c_1 \exp(-c_2 n^{1/3})$. 2) $|\hat{\mathbf{X}}_{ij}| \leqslant r^{2/3}$ for all $i, j$ via Lemma 1. 3) $|Y(\lambda) - \mathbb{E}[y(\lambda)]| \leqslant cr^{2/3}n^2$ for 3 evenly spaced $\lambda_i$ by the above equation.

Now, since the $\ell''(\lambda)$ is a second-degree polynomial, if it's close to its expectation at 3 places, it must be close it its expectation everywhere. Formally, we can define the second-degree polynomial $\hat{y}(\lambda) = y(\lambda) - \mathbb{E}[y(\lambda)]$. Now, our union bounds states that $|\hat{y}(\lambda_i)| < cr^{2/3}n^2$ for three evenly spaced $\lambda_i$. Now, let the three coefficients of the second-degree polynomial be described by the vector $\mathbf{p}$, its values at $\lambda_i$ by the vector $\mathbf{v}$ and let the matrix $\mathbf{A}$ describe the values of the monomials $1, \lambda, \lambda^2$ at $\lambda_i$. We then have $\mathbf{Ap} = \mathbf{v}$. Now, $\mathbf{A}$ will have elements bounded by 1 but will be full rank, this implies that if $\mathbf{v}$ is bounded by $cr^{2/3}n^2$ so must $\mathbf{p}$ be. The fact that the coefficents in $\hat{y}$ have absolute value bounded by $\mathcal{O}(r^{2/3}n^2)$, together with the fact that the mean scales as $rn^2$ gives us the following

$$\min_\lambda y(\lambda) = \min_\lambda \hat{y}(\lambda) + \mathbb{E}[y(\lambda)] \geqslant \min_\lambda \hat{y}(\lambda) + \min_{\lambda'} \mathbb{E}[y(\lambda')]$$

$$\geqslant \min_\lambda \hat{y}(\lambda) + crn^2 \geqslant c' r^{2/3}n^2 + crn^2$$

Here $c$ is positive as per our assumptions, whereas $c'$ might be negative. Now, for sufficiently large $r$, the above quality will be non-negative. We can thus find new constants, based upon $c$ and $c'$, to complete the proof. ∎

## D.6 Concentration

This section contains the concentration results needed for the proof of Theorem 1. We will use results from random matrix theory, which has stronger results for matrices with independent entries that are concentrated (Ahlswede and Winter, 2002; Tropp, 2012; Guionnet et al., 2000). We will use the concentration results of Meckes and Szarek (2012) which relies on the notation of CCP – convex concentration property – which is a regularity condition.

**Definition 2.** *A random matrix $\mathbf{X}$ in a normed vector space satisfies CCP iff there exists positive constants $c_1, c_2$ such that $P\left(\left|f(\mathbf{X}) - \mathbb{M}f(\mathbf{X})\right| \geqslant t\right) \leqslant c_1 e^{-c_2 t^2}$ for all $t$ and all convex 1-Lipschitz functions $f$. Here $\mathbb{M}$ is the median.*

One can easily verify that independent Gaussian variables satisfies this (Ledoux, 2001), and it then follows that 1-Lipschitz functions of Gaussians also satisfies CCP. One important class here is the half-normal distribution which is just the absolute value of a Gaussuan variable, and absolute value is a 1-Lipschitz function. Any vector of independent bounded variables will also satisfy CCP (Meckes and Szarek, 2012).

**Fact 1.** *Let $\hat{P}$ be a polynomial of degree 4 in centered matrices $\mathbf{X}_i, \mathbf{X}_j, \mathbf{X}_k, \mathbf{X}_l$ otherwise sampled as per Assumption 1, where we allow $i = j$, $j = l$ and any other arbitrary index relationship. Let $z = \mathrm{Tr}\,\hat{P}(\mathbf{X}_i, \mathbf{X}_j, \mathbf{X}_k, \mathbf{X}_l)$ and let $\mu$ be then mean of $z$, then $P\left(|z - \mu| > t\,r\,n^2\right) \leqslant c_1 \exp\left(-c_2 \min(t^2, t^{1/2})\,n\right)$ for positive constants $c_1, c_2$.*

**Proof.** Any vector CCP where some components are zero will still be CCP, this we can pad the matrices $\mathbf{X}_i, \mathbf{X}_j, \mathbf{X}_k, \mathbf{X}_l$ so that they are square. We then invoke Theorem 1 of Meckes and Szarek (2012) which gives

$$P\left(|z - \mu| > \tau n^2\right) \leqslant c_1 \exp\left(-c_2 \min(\tau^2, n\,\tau^{1/2})\right)$$

Setting $\tau = tr$ gives us

$$P\left(|z - \mu| > t\,r\,n^2\right) \leqslant c_1 \exp\left(-c_2 \min(t^2\,r^2,\, t^{1/2}\,r^{1/2}\,n)\right)$$

Using our assumption $r = \mathcal{O}(n^\gamma)$ for some $\gamma \in [1/2, 1]$ gives

$$P\left(|z - \mu| > t\,r\,n^2\right) \leqslant c_1 \exp\left(-c_2 \min(t^2, t^{1/2})\,n\right) \qquad \blacksquare$$

**Fact 2.** *Let $z = \mathrm{Tr}\left(\mathbf{1}\mathbf{X}_i\mathbf{X}_j\mathbf{X}_k\right)$ for centered matrices $\mathbf{X}_i, \mathbf{X}_j, \mathbf{X}_k$ otherwise sampled as per Assumption 1, and let $\mu$ be the mean of $z$. $\mathbf{1}$ is a matrix of all 1s of the appropriate shape. Then $P\left(|z - \mu| \geqslant t\,rn^2\right) \leqslant c_1 \exp\left(-c_2 n^{1/3} \min(t^{1/2}, t^2)\right)$ for positive constants $c_1, c_2$.*

**Proof.** As before, we can pad the matrices to be square while retaining the CCP property. We note that the singular value of the matrix $\mathbf{1}$ is at most $n$. This means that the shatten-norm $\|\mathbb{E}\,\mathbf{1}\|_k$ is at most $n$ for all $k$. Let us consider the matrix $\hat{\mathbf{1}} = \frac{1}{n^{1/3}}$ which then has shatten-norm $\|\mathbb{E}\,\hat{\mathbf{1}}\|_d$ is at most $n^{2/3}$. For this matrix, Theorem 1 in Meckes and Szarek (2012) applies, so that we have

$$P\left[\left|\mathrm{Tr}\left(\mathbf{1}\mathbf{X}_i\mathbf{X}_j\mathbf{X}_k - \mu\right)\right| \geqslant tn^2\right] \leqslant c_1 \exp\left(-c_2 \min\left(t^2, nt^{1/2}\right)\right) \tag{12}$$

Let us take $t = t_0 r n^{-1/3}$ so that $tn^{7/3} = t_0 rn^2$. We then have

$$\left|\mathrm{Tr}\left(\hat{\mathbf{1}}\mathbf{X}_i\mathbf{X}_j\mathbf{X}_k - \mu\right)\right| \geqslant tn^2 \iff \left|\mathrm{Tr}\left(\mathbf{1}\mathbf{X}_i\mathbf{X}_j\mathbf{X}_k - \mu\right)\right| \geqslant tn^{7/3}$$

$$\iff \left|\mathrm{Tr}\left(\mathbf{1}\mathbf{X}_i\mathbf{X}_j\mathbf{X}_k - \mu\right)\right| \geqslant t_0 rn^2$$

By substituting $t = t_0 r n^{-1/3}$ into equation 12 we then have

$$P\left(\left|\mathrm{Tr}\left(\mathbf{1}\,\mathbf{X}_i\mathbf{X}_j\mathbf{X}_k - \mu\right)\right| \geqslant t_0 rn^2\right) \tag{13}$$

$$\leqslant c_1 \exp\left(-c_2 \min\left(t_0^2 r^2 n^{-2/3},\, t_0^{1/2} r^{1/2} n^{1-1/6}\right)\right)$$

Recall our assumption $r = c_3 n^\gamma$ for $\gamma \in [1/2, 1]$. We then have

$$\min\left(t_0^2 r^2 n^{-2/3},\, t_0^{1/2} r^{1/2} n^{1-1/6}\right) \geqslant \min\left(t_0^{1/2}, t_0^2\right) n^{1/3}$$

Plugging this into equation 13 completes the proof $\blacksquare$

**Lemma 1.** *With probability $\leqslant c_1 n^2 \exp\left(-c_2 r^{1/3}\right)$, no entry in $\hat{\mathbf{X}}_{ij}, \hat{\mathbf{X}}'_{ij}, \hat{\mathbf{X}}''_{ij}, \mathbf{X}_{ij}$ has an absolute value larger than $cr^{2/3}$, for some positive constant $c$.*

**Proof.** For a fixed $i, j$ consider the terms $\hat{\mathbf{X}}_{ij}, \hat{\mathbf{X}}'_{ij}, \hat{\mathbf{X}}''_{ij}, \mathbf{X}_{ij}$. Fact 9 says that each one can be expressed as a constant number of variables $a$ with zero mean of the form

$$a = \sum_{i=1}^{r} \mathbf{v}_i^{(1)} \mathbf{v}_i^{(2)}$$

Since the variables themselves are sub-gaussian, by Lemma 2.7.7 in Vershynin (2018) the product $\mathbf{v}_i^{(1)} \mathbf{v}_i^{(2)}$ is sub-exponential. Thus, the variable $a$ is a sum of iid sub-exponential variables, and Theorem 2.8.1 in Vershynin (2018) states that

$$P\left(|a| \geqslant t\right) \leqslant 2 \exp\left(-c \min\left[\frac{t^2}{rK^2}, \frac{t}{K}\right]\right)$$

Here $K$ is the orlicz norm $\| \cdot \|_{\Psi_1}$ which is a constant for our fixed distributions. In the above expression, let us set $t = cr^{2/3}$. This gives

$$P\left( \left| a - \mathbb{E}[a] \right| \geqslant cr^{2/3} \right) \leqslant 2 \exp\left( -cr^{1/3} \right)$$

We note that there is a polynomial number of entries $a$, so we can do a polynomial number (w.r.t. $n$) of union bounds, which might change the constants, to show that no entry is further than $c_1 r^{2/3}$ from its expectation for some positive $c_1$. ∎

# E  ALGEBRAIC CALCULATIONS

## E.1  BASIC FACTS

**Fact 3.** $\mathbb{E}\left[ \operatorname{Tr} \bar{\mathbf{U}}^T \bar{\mathbf{U}} \bar{\mathbf{V}} \bar{\mathbf{V}}^T \right] = rmn\sigma^4$

**Proof.** We have

$$\mathbb{E}\left[ \operatorname{Tr} \bar{\mathbf{U}}^T \bar{\mathbf{U}} \bar{\mathbf{V}} \bar{\mathbf{V}}^T \right] = \mathbb{E}\left[ \sum_{ijkl} \bar{\mathbf{U}}_{ji} \bar{\mathbf{U}}_{jk} \bar{\mathbf{V}}_{kl} \bar{\mathbf{V}}_{il} \right]$$

Taking the mean, and using linearity of expectations, we only have nonzero mean if $i = k$. We get

$$= \mathbb{E}\left[ \sum_{ijl} \bar{\mathbf{U}}_{ji} \bar{\mathbf{U}}_{ji} \bar{\mathbf{V}}_{il} \bar{V}_{il} \right]$$

Using linearity of expectation and independence, this becomes

$$\sum_{ijl} \mathbb{E}\left[ \bar{\mathbf{U}}_{ji} \bar{\mathbf{U}}_{ji} \right] \mathbb{E}\left[ \bar{\mathbf{V}}_{il} \bar{\mathbf{V}}_{il} \right]$$

For a single matrix entry, $\mathbb{E}\left[ \bar{\mathbf{U}}_{ji} \bar{\mathbf{U}}_{ji} \right] = \sigma^2$. $i$ goes over the columns of $\mathbf{U}$ of which there are $r$, $j$ goes over the rows of $\bar{\mathbf{U}}$ of which there are $n$ and $l$ goes over the columns of $\bar{\mathbf{V}}$ of which there are $m$, so we clearly get $\sigma^4 rmn$. ∎

**Fact 4.** $\mathbb{E}\left[ \operatorname{Tr} \bar{\mathbf{U}}^T \bar{\mathbf{U}} \boldsymbol{I}_{r \times m} \boldsymbol{I}_{r \times m}^T \right] = rmn\mu_{var}^2 \sigma^2$

**Proof.** We can can first contract $\mathbf{1}_{r \times m} \mathbf{1}_{r \times m}^T = \mathbf{1}m\mu_{var}^2$, where $1$ is a $r$-by-$r$ matrix of all ones. We then want to find

$$\mathbb{E}\left[ \sum_{ijk} \bar{\mathbf{U}}_{ji} \bar{\mathbf{U}}_{jk} \mathbf{1}_{ki} \right]$$

Now since the variables are centered, the expectations becomes zero unless $i = k$. The sum thus becomes

$$\mathbb{E}\left[ \sum_{ij} \bar{\mathbf{U}}_{ji} \bar{\mathbf{U}}_{ji} \mathbf{1}_{ii} \right] = nr\sigma^2$$

This gives us the result of $rmn\mu_{var}^2 \sigma^2$. ∎

## E.2  EXPECTATION

**Fact 5.** $\mathbb{E}[\|\mathbf{W}_2\|^2] = 4mnr\sigma^4$

**Proof.** Let us rewrite $\mathbb{E}[\|\mathbf{W}_2\|^2]$ as

$$\mathbb{E}\operatorname{Tr}\left[\left(\mathbf{V}_1 - \mathbf{V}_2\right)^T \left(\mathbf{U}_1 - \mathbf{U}_2\right)^T \left(\mathbf{U}_1 - \mathbf{U}_2\right)\left(\mathbf{V}_1 - \mathbf{V}_2\right)\right]$$

Trace and expectations are linear operators, so we can reorder them. Let us consider the matrix

$$\left(\mathbf{V}_1 - \mathbf{V}_2\right)^T \left(\mathbf{U}_1 - \mathbf{U}_2\right)^T \left(\mathbf{U}_1 - \mathbf{U}_2\right)\left(\mathbf{V}_1 - \mathbf{V}_2\right)$$

We can easily add and subtract the mean $\mu$ of all matrices which gives

$$\left(\bar{\mathbf{V}}_1 - \bar{\mathbf{V}}_2\right)^T \left(\bar{\mathbf{U}}_1 - \bar{\mathbf{U}}_2\right)^T \left(\bar{\mathbf{U}}_1 - \bar{\mathbf{U}}_2\right)\left(\bar{\mathbf{V}}_1 - \bar{\mathbf{V}}_2\right) \tag{14}$$

Expanding the parantheses gives us 16 matrices, however they will have zero mean unless both the $\mathbf{U}$-matrices and $\mathbf{V}$-matrices ar the same. For calculating the mean, we can thus only consider four matrices of the type $\mathbf{V}^T\mathbf{U}^T\mathbf{U}\mathbf{V}$. Permuting the indices cyclically and appealing to Fact 3 gives us

$$\mathbb{E}[\|\mathbf{W}_2\|^2] = 4mnr\sigma^4 \qquad \blacksquare$$

**Fact 6.** $\mathbb{E}[\langle\mathbf{W}_0, \mathbf{W}_2\rangle] = rmn\sigma^4$

**Proof.** We can write $\mathbb{E}[\langle\mathbf{W}_0, \mathbf{W}_2\rangle]$ as

$$\mathbb{E}\operatorname{Tr}\left[\left(\mathbf{V}_2^T\mathbf{U}_2^T - (\mathbf{V}^*)^T(\mathbf{U}^*)^T\right)\left((\mathbf{U}_1 - \mathbf{U}_2)(\mathbf{V}_1 - \mathbf{V}_2)\right)\right]$$

Again, we want to add and subtract the matrix means $\mathbf{1}_{n\times r}, \mathbf{1}_{r\times m}$ to get

$$\mathbb{E}\operatorname{Tr}\left[\left(\left[\bar{\mathbf{V}}_2^T\bar{\mathbf{U}}_2^T + \bar{\mathbf{V}}_2^T\mathbf{1}_{n\times r}^T + \mathbf{1}_{r\times m}^T\bar{\mathbf{U}}_2^T + \mathbf{1}_{r\times m}^T\mathbf{1}_{n\times r}^T\right] - \left[\bar{\mathbf{V}}^{*T}\bar{\mathbf{U}}^{*T} + \bar{\mathbf{V}}^{*T}\mathbf{1}_{n\times r}^T + \mathbf{1}_{r\times m}^T\bar{\mathbf{U}}^{*T} + \mathbf{1}_{r\times m}^T\mathbf{1}_{n\times r}^T\right]\right)\right.$$

$$\left.\times\left(\left(\bar{\mathbf{U}}_1 - \bar{\mathbf{U}}_2\right)\left(\bar{\mathbf{V}}_1 - \bar{\mathbf{V}}_2\right)\right)\right] \tag{15}$$

As before, we can remove terms which are linear in any centered variable. This removes matrices with index $1, 3$ and leaves

$$\mathbb{E}\operatorname{Tr}\left[\left(\bar{\mathbf{V}}_2^T\bar{\mathbf{U}}_2^T + \bar{\mathbf{V}}_2^T\mathbf{1}_{n\times r}^T + \mathbf{1}_{r\times m}^T\bar{\mathbf{U}}_2^T\right)\left(\bar{\mathbf{U}}_2\bar{\mathbf{V}}_2\right)\right]$$

Applying fact 3 to the only term that remain gives

$$\mathbb{E}[\langle\mathbf{W}_0, \mathbf{W}_2\rangle] = \mathbb{E}\operatorname{Tr}\left[\bar{\mathbf{V}}_2^T\bar{\mathbf{U}}_2^T\bar{\mathbf{U}}_2\bar{\mathbf{V}}_2\right] = rmn\sigma^4 \qquad \blacksquare$$

**Fact 7.** $\mathbb{E}\|\mathbf{W}_1\|_F^2 = 6rmn\sigma^4 + 4rmn\mu_{var}^2\sigma^2$

**Proof.** We rewrite $\mathbb{E}\|\mathbf{W}_1\|_F^2$ as

$$\mathbb{E}\operatorname{Tr}\left[\left(\mathbf{V}_2^T\left(\mathbf{U}_1 - \mathbf{U}_2\right)^T + \left(\mathbf{V}_1 - \mathbf{V}_2\right)^T\mathbf{U}_2^T\right)\left(\left(\mathbf{U}_1 - \mathbf{U}_2\right)\mathbf{V}_2 + \mathbf{U}_2\left(\mathbf{V}_1 - \mathbf{V}_2\right)\right)\right]$$

We will again want to center the variables, which gives

$$\mathbb{E}\operatorname{Tr}\left[\left(\bar{\mathbf{V}}_2^T\left(\bar{\mathbf{U}}_1 - \bar{\mathbf{U}}_2\right)^T + \mathbf{1}_{r\times m}^T\left(\bar{\mathbf{U}}_1 - \bar{\mathbf{U}}_2\right)^T + \left(\bar{\mathbf{U}}_1 - \bar{\mathbf{V}}_2\right)^T\bar{\mathbf{U}}_2^T + \left(\bar{\mathbf{V}}_1 - \bar{\mathbf{V}}_2\right)^T\mathbf{1}_{n\times r}^T\right)\right.$$

$$\times \left( \left( \bar{\mathbf{U}}_1 - \bar{\mathbf{U}}_2 \right) \bar{\mathbf{V}}_2 + \left( \bar{\mathbf{U}}_1 - \bar{\mathbf{U}}_2 \right) \mathbf{1}_{r \times m} + \bar{\mathbf{U}}_2 \left( \bar{\mathbf{V}}_1 - \bar{\mathbf{V}}_2 \right) + \mathbf{1}_{n \times r} \left( \bar{\mathbf{V}}_1 - \bar{\mathbf{V}}_2 \right) \right) \right]$$

Let us first consider the terms involving constants. We remove any terms linear in centered variables. We are then left with

$$\mathbf{1}_{r \times m}^T \left( \bar{\mathbf{U}}_1 - \bar{\mathbf{U}}_2 \right)^T \left( \bar{\mathbf{U}}_1 - \bar{\mathbf{U}}_2 \right) \mathbf{1}_{r \times m} + \left( \bar{\mathbf{V}}_1 - \bar{\mathbf{V}}_2 \right)^T \mathbf{1}_{n \times r}^T \mathbf{1}_{n \times r} \left( \bar{\mathbf{V}}_1 - \bar{\mathbf{V}}_2 \right) \tag{16}$$

As before, any expressions linear in centered variables vanish when we take expectations. Using this and Fact 4 we get that the above expression is equal to

$$= 4rmn\mu_{var}^2 \sigma^2$$

We now return to the terms without constants These are

$$\bar{\mathbf{V}}_2^T \left( \bar{\mathbf{U}}_1 - \bar{\mathbf{U}}_2 \right)^T \left( \bar{\mathbf{U}}_1 - \bar{\mathbf{U}}_2 \right) \bar{\mathbf{V}}_2 + \left( \bar{\mathbf{V}}_1 - \bar{\mathbf{V}}_2 \right)^T \bar{\mathbf{U}}_2^T \bar{\mathbf{U}}_2 \left( \bar{\mathbf{V}}_1 - \bar{\mathbf{V}}_2 \right) \tag{17}$$

$$+ \bar{\mathbf{V}}_2^T \left( \bar{\mathbf{U}}_1 - \bar{\mathbf{U}}_2 \right)^T \bar{\mathbf{U}}_2 \left( \bar{\mathbf{V}}_1 - \bar{\mathbf{V}}_2 \right) + \left( \bar{\mathbf{V}}_1 - \bar{\mathbf{V}}_2 \right)^T \bar{\mathbf{U}}_2^T \left( \bar{\mathbf{U}}_1 - \bar{\mathbf{U}}_2 \right) \bar{\mathbf{V}}_2$$

As before, we remove any expressions linear in centered variables. As it turns out, we are left with 6 terms of the type $\bar{\mathbf{V}}_2^T \bar{\mathbf{U}}_2^T \bar{\mathbf{U}}_2 \bar{\mathbf{V}}_2$. Applying Fact 3 gives us

$$\mathbb{E}\big[ \|\mathbf{W}_1\|_F^2 \big] = 6rmn\sigma^4 + 4rmn\mu_{var}^2 \sigma^2 \qquad \blacksquare$$

**Fact 8.** $\mathbb{E}\langle \mathbf{W}_2, \mathbf{W}_1 \rangle = -4rmn\sigma^4$

**Proof.** Let us rewrite $\mathbb{E}\langle \mathbf{W}_2, \mathbf{W}_1 \rangle$ as

$$\mathbb{E} \operatorname{Tr} \left[ \left( \left( \mathbf{V}_1 - \mathbf{V}_2 \right)^T \left( \mathbf{U}_1 - \mathbf{U}_2 \right)^T \right) \left( \left( \mathbf{U}_1 - \mathbf{U}_2 \right) \mathbf{V}_2 + \mathbf{U}_2 \left( \mathbf{V}_1 - \mathbf{V}_2 \right) \right) \right]$$

As per usual, we center the variables

$$= \left[ \left( \left( \bar{\mathbf{V}}_1 - \bar{\mathbf{V}}_2 \right)^T \left( \bar{\mathbf{U}}_1 - \bar{\mathbf{U}}_2 \right)^T \right) \left( \left( \bar{\mathbf{U}}_1 - \bar{\mathbf{U}}_2 \right) \bar{\mathbf{V}}_2 + \left( \bar{\mathbf{U}}_1 - \bar{\mathbf{U}}_2 \right) \mathbf{1}_{r \times m} + \bar{\mathbf{U}}_2 \left( \bar{\mathbf{V}}_1 - \bar{\mathbf{V}}_2 \right) + \mathbf{1}_{n \times r} \left( \bar{\mathbf{V}}_1 - \bar{\mathbf{V}}_2 \right) \right) \right] \tag{18}$$

Any terms involving the constant will be linear in a centered variable, and thus disappear under expectations. Equating like terms and using Fact 3, we get

$$= -4\mathbb{E}\left[ \operatorname{Tr} \bar{\mathbf{V}}_1^T \bar{\mathbf{U}}_1^T \bar{\mathbf{U}}_1 \bar{\mathbf{V}}_1 \right] = -4rmn\sigma^4 \qquad \blacksquare$$

### E.3 Other algebraic facts

**Fact 9.** *For any fixed $i, j$ the entries of $\hat{\mathbf{X}}, \hat{\mathbf{X}}'$ and $\hat{\mathbf{X}}''$ are zero mean.*

**Proof.** We have

$$\hat{\mathbf{X}}(\lambda) - \mathbf{X} = \left( \lambda \mathbf{U}_1 + (1 - \lambda)\mathbf{U}_2 \right) \left( \lambda \mathbf{V}_1 + (1 - \lambda)\mathbf{V}_2 \right) - \mathbf{U}^* \mathbf{V}^*$$

$$\hat{\mathbf{X}}'(\lambda) = \left( \mathbf{U}_1 - \mathbf{U}_2 \right) \left( \lambda \mathbf{V}_1 + (1 - \lambda)\mathbf{V}_2 \right) + \left( \lambda \mathbf{U}_1 + (1 - \lambda)\mathbf{U}_2 \right) \left( \mathbf{V}_1 - \mathbf{V}_2 \right)$$

$$\hat{\mathbf{X}}''(\lambda) = \left( \mathbf{U}_1 - \mathbf{U}_2 \right) \left( \mathbf{V}_1 - \mathbf{V}_2 \right)$$

The fact that $\mathbf{U}_1$ and $\mathbf{U}_2$ are iid implies that $(\mathbf{U}_1 - \mathbf{U}_2)$ has zero mean, and the similar holds for $\mathbf{V}_1$ and $\mathbf{V}_2$. This implies that all terms of $\hat{\mathbf{X}}'$ and $\hat{\mathbf{X}}''$ are zero-mean. Using the fact that $\mathbf{U}_1, \mathbf{U}_2$ and $\mathbf{U}^*$ are iid implies that

$$\mathbb{E}\left[\left(\lambda\mathbf{U}_1 + (1-\lambda)\mathbf{U}_2\right)\left(\lambda\mathbf{V}_1 + (1-\lambda)\mathbf{V}_2\right)\right] = \mathbb{E}[\mathbf{U}^*\mathbf{V}^*]$$

This, in turn, implies that $\mathbb{E}[\hat{\mathbf{X}}] = 0$. $\blacksquare$

**Fact 10.** *For* $\mathbf{U}_1 = \mathbf{U}^*$ *and* $\mathbf{V}_1 = \mathbf{V}^*$, *we have* $\mathbb{E}[\langle\mathbf{W}_0, \mathbf{W}_2\rangle] = 2rmn\sigma^4$

**Proof.** We can write $\mathbb{E}[\langle\mathbf{W}_0, \mathbf{W}_2\rangle]$ as

$$\mathbb{E}\,\mathrm{Tr}\left[\left(\mathbf{V}_2^T\mathbf{U}_2^T - (\mathbf{V}^*)^T(\mathbf{U}^*)^T\right)\left((\mathbf{U}^* - \mathbf{U}_2)(\mathbf{V}^* - \mathbf{V}_2)\right)\right]$$

Again, we want to add and subtract the matrix means $\mathbf{1}_{n\times r}$, $\mathbf{1}_{r\times m}$ to get

$$\mathbb{E}\,\mathrm{Tr}\left[\left(\left[\bar{\mathbf{V}}_2^T\bar{\mathbf{U}}_2^T + \bar{\mathbf{V}}_2^T\mathbf{1}_{n\times r}^T + \mathbf{1}_{r\times m}^T\bar{\mathbf{U}}_2^T + \mathbf{1}_{r\times m}^T\mathbf{1}_{n\times r}^T\right] - \left[\bar{\mathbf{V}}^{*T}\bar{\mathbf{U}}^{*T} + \bar{\mathbf{V}}^{*T}\mathbf{1}_{n\times r}^T + \mathbf{1}_{r\times m}^T\bar{\mathbf{U}}^{*T} + \mathbf{1}_{r\times m}^T\mathbf{1}_{n\times r}^T\right]\right)\right.$$

$$\left. \times\left((\bar{\mathbf{U}}^* - \bar{\mathbf{U}}_2)(\bar{\mathbf{V}}^* - \bar{\mathbf{V}}_2)\right)\right] \tag{19}$$

As before, we can remove terms which are linear in any centered variable. The only term that remain gives

$$\mathbb{E}[\langle\mathbf{W}_0, \mathbf{W}_2\rangle] = \mathbb{E}\,\mathrm{Tr}\left[\bar{\mathbf{V}}_2^T\bar{\mathbf{U}}_2^T\bar{\mathbf{U}}_2\bar{\mathbf{V}}_2\right] + \mathbb{E}\,\mathrm{Tr}\left[\bar{\mathbf{V}}^{*T}\bar{\mathbf{U}}^{*T}\bar{\mathbf{U}}^*\bar{\mathbf{V}}^*\right] = 2rmn\sigma^4 \qquad \blacksquare$$

