# OpenReview forum: "Star-Convexity in Non-Negative Matrix Factorization"
_ICLR.cc/2020/Conference — Reject_

### Official Review · AnonReviewer1 · 2019-10-19
**Official Blind Review #1**

**Rating:** 3

**Review:**

The paper derives results for nonnegative-matrix factorization along the lines of recent results on SGD for DNNs, showing that the loss is star-convex towards randomized planted solutions. The star-convexity property is also shown to hold to some degree on real world datasets. The paper argues that these results explain the good performance that usual gradient descent procedures achieve in practice. The paper also puts forward a conjecture that more parameters make the loss function easier to optimize by making it more likely that star convexity holds, and that a similar conclusion could hold for DNNs.

The paper is rather well written, although there are many small typos or notation errors (of which I mention a few below). In addition, I have a few issues with the presentation of both the theoretical and experimental results. Although the results relating to star convexity seem compelling and interesting to understand the good practical performance of the usual simple NMF algorithms, I find the conjecture on concentration of measure a bit hand-wavy.

In particular, I have some questions regarding the main theorem:

	- "r grows as $O(n^\gamma)$ and m as $O(n)$": Big-O means asymptotically bounded, i.e., that r is roughly smaller than n^\gamma (which is a tautology for $\gamma=1$ anyway), and that m is roughly smaller than n. Is that really what you mean? Or do you mean $\Omega/\Theta$ instead? Fact 2 uses "recall our assumption that $r = c n^\gamma$" which seems to imply you mean "r = \Omega(n^\gamma)", which can also be informally stated as "r grows as n^\gamma", without Big-O notation. Please clarify this notation.

	- Also, re $\gamma \geq 0.5$, I do not quite see why such a strong assumption is made. Wouldn't $r = \Omega(n^{1/3 + \varepsilon})$ be enough with respect to Fact 2 to get an asymptotically vanishing deviation probability? i.e., wouldn't the final result work with $\gamma > 1/3`$, which is more general?

Generally, this begs the question of whether $r = \Omega(n^0.5)$ is realistic. Can this be put in perspective with respect to other work?

	- What is Lemma 9 mentioned in the proof of Lemma 1? Is that in the paper? I could not find it. In particular, I am unclear on "we can do a polynomial number of union bounds": polynomial with respect to which variable?

I also have the following additional questions on the experimental findings and the proposed conjecture.

- Section 4.2 and Figure 5 uses the 'relative deviation', which is the standard deviation normalized by the mean. I am not sure what conclusion to draw from this, however. In particular:
  - Why normalize by dividing by the mean?
  - What is the evolution of the mean itself? If the mean is always greater than a few standard deviations, then the fact that curvature is getting more concentrated may not matter. How about showing instead the fraction of trials where the curvature was non-negative? (i.e., where we had convexity along the line)

- I struggle a bit to make sense of the conjecture. In particular, 'concentration of measure' is related to randomness, but it isn't quite clear which 'measure' we're talking about here. Table 2 is also not very convincing: although the legend indicates 'increased width makes the loss surface increasingly locally convex', it only seems to hold for the fourth row (epoch=300). In addition, these means are reported without standard deviations, making it hard to judge whether to trust the ordering of these few numbers.

Typos/unclear:

introduction: "strictly non-negative factors", what do you mean by strictly?
"randomly chosen or the global minimizer" is a bit unclear - what is the 'or' over?
"Convex along straight paths towards the optima x*": x* -> $x^*$ to be consistent with Definition 1.
"similar to Dvoretzky's": I fail to see the similarity and how this is related to the present paper.
"we will assume that there is a planted optimal solution (U*, V*)": U*, V* should be in bold to be consistent with further notation.
"how r and m depends on n": depends -> depend
"as the size of the problem increaseS"
"loss function *of* equation 2"
Theorem 1: "at least \geq" is redundant
Theorem 1: "but with exponent -c r^{1/3}" is unclear: which exponent is this? Write the statement in full instead.
"it's second derivative": its?
"for unobserved Data": why is 'data' capitalized?
D3: "clearly, equation 9 holds in this case": not sure what is meant here - equation (10) does not imply (9) or reciprocally. Did you mean that equation 10 holds instead?
D3: "3 evenly spaceD \lambda"
D6: Definition 2: "iff there *exists* positive constants..."
D6: "one can easily verify that independenT Gaussian variables"
D6: "i=j, j=l, and so on": what is 'and so on'? all indexes are completely arbitrary?
D6: "z = P(X_i, X_j, X_k, X_l)": clarify notation $P$. In the main text, you've used the notation $p$ instead.
D6: Equation (12) and (13): should the RHS be exp(-X) instead of exp(X)?
D6: $- \mu$ notation is a bit confusing: the text doesn't explain what it is, and lack of bold face hints that this is a scalar and not a matrix.
D6: "Lemma 9 says that each one can be expressED"
E1: End of proof of fact 3: do not include an equal sign if there is no left-hand side.
E1: "trace and linearity are linear operators": linearity is linear?


**Experience Assessment:**

I have read many papers in this area.

**Review Assessment: Checking Correctness Of Derivations And Theory:**

I assessed the sensibility of the derivations and theory.

**Review Assessment: Checking Correctness Of Experiments:**

I assessed the sensibility of the experiments.

**Review Assessment: Thoroughness In Paper Reading:**

I read the paper thoroughly.

---

> ### Author Response · Authors · 2019-11-12
> **Thanks for the comments**
>
> We thank the Reviewer for their thoughtful comments and constructive criticism. We are happy that the reviewer thinks that the idea of star convexity is compelling and that it could explain the practical performance of NMF.
>
> We did not mean to make a theoretical conjecture on the concentration of measure, but more point toward it as a general phenomenon. In our theorem you can see that for larger number of dimensions one gets better probability bound; our aim was to say that this might be a general phenomenon.
>
> The reviewer is correct in pointing out that our main bound can be tightened. We did not do this to make the theorem seem clean. But it would indeed be a good idea to mention that one can improve the constants slightly at the cost of a more complicated expression.
>
> We are not aware of any other work which shows asymptotic dimensions for non-negative matrix factorization. For some data sets we certainly think it’s realistic, but some data sets are definitely sparser.
>
> Regarding smaller comments:
> The reason why we wanted to normalize by the mean is to remove the effect of a uniform scaling. During our experiments there were never any trials when the curvature was non-negative. For this reason, we did not show the fraction, and thought that standard deviation over mean was appropriate. Regarding Table 2, the results are indeed strongest for the fourth row which is epoch 300. We were not able to add standard deviations simply because the page width would not allow it. The big-O notation is indeed used misleadingly, we meant to say that r grows roughly as n to the gamma.
>
> We thank the Reviewer for pointing out the typos and where clarifications are needed.

---

> > ### Author Response · Authors · 2019-11-12
> > **changes made**
> >
> > * we have clarified how r grows.
> > * we have made a clarification that one can get slightly stronger results at the cost of a more complicated expression.
> > * we have changed “conjecture” to “hypothesise” and have clarified it.
> > * “lemma 9” was supposed to be “fact 9”. We have fixed this typo.
> > * we have fixed the typos and clarifications requested.

---

### Official Review · AnonReviewer3 · 2019-10-22
**Official Blind Review #3**

**Rating:** 6

**Review:**

1.	Typos:
(1)	“We will consider slighly weaker notation of…” in page 3 should be “We will consider slightly weaker notation of…”.
(2)	“The same holds along the line line…” in page 3 should be “The same holds along the line…”.
(3)	A comma should be added between $U^*$ and $V_1$ in the Subsection ‘Proof sketch of unobserved Data’ in page 5.
2.	Figure 3 presents the loss surface of NMF on straight paths connecting two random points for 8 real-world datasets. From this figure, we can see that the minima is obtained around lambda equals to 0.5. Whether the authors can explain this phenomenon? Besides, the authors should annotate the differences among each curve in each subfigure in Figure 3 and Figure 4.
3.	Experiments utilize 8 datasets to demonstrate the good performance of the proposed planted model. The decomposition rank are given in the previously literature. However, in practice, the rank of a new dataset is unavilable. How to handel this situation?
4.	There are unknow elements in Goodbooks, Movielens and Netflix datasets. These can be processed by the given sparsity. However, there is no items in the proposed planted model that can handle the sparsity.
5.     Open codes about this paper.


**Experience Assessment:**

I have read many papers in this area.

**Review Assessment: Checking Correctness Of Derivations And Theory:**

I assessed the sensibility of the derivations and theory.

**Review Assessment: Checking Correctness Of Experiments:**

I did not assess the experiments.

**Review Assessment: Thoroughness In Paper Reading:**

I read the paper at least twice and used my best judgement in assessing the paper.

---

> ### Author Response · Authors · 2019-11-12
> **Thanks for the comments**
>
> We thank Reviewer 2 for his constructive comments and thorough review.
>
> The fact that the loss is convex between the two points is in accordance with our theoretical results. The fact that it seems to be minimized exactly at 0.5 is not accounted for in our model, but qualitatively it is as our theory predicts.
>
> The figures could indeed be annotated, thank you for this suggestion.
>
> In cases when the rank is unavailable, a common strategy is to try different ranks and see for what rank diminishing returns kick in.
>
> Our theoretical model handles sparsity in the sense that we can have data that is sparsely observed from a dense matrix. In this case, the data is observed independently with some fixed small probability.
>
> We can open source the code base upon publication.
>
> Finally, we thank the reviewer for pointing out all the typos.

---

> > ### Author Response · Authors · 2019-11-12
> > **changes made**
> >
> > we have fixed the typos

---

### Official Review · AnonReviewer2 · 2019-10-22
**Official Blind Review #2**

**Rating:** 3

**Review:**

This paper studies loss landscape of Non-negative matrix factorization (NMF) when the matrix is very large. It shows that with high probability, the landscape is quasi-convex under some conditions. This suggests that the optimization problem would become easier as the size of the matrix becomes very large. Implications on deep networks are also discussed.

The NMF problem is known to be NP-hard. In case that the matrix X to factorize is large, the author(s) uses concentration property of random matrix to show that along any random positive matrix U,V and U’,V’, the MSE loss of NMF is convex with high probability.  The extra assumption is that the rank of U and V should also be large enough. Section 3 is devoted to prove this. It seems to me there are some typos which are quite serious and make the equation (3) incorrect. However, the main result (Theorem 1) still seems to hold. The equation (3) should replace W_2 with 2 W_2. The reason is in the appendix D.2, the definition of W_2 has missed this constant 2, which is the hat X’’(lambda) at lambda=0. Therefore, all the constants in the equation (4) need to be modified accordingly. In D.2, to derive the equation (9), it seems to me the McLaurin series should give 2 l’’’’(0) l’’(0) >= (l’’’(0))^2, isn’t it? The whole proof is quite long to check. In Fact 1, is the mu the mean of z? In Lemma 1, what is lemma 9? Fact 10 has a constant 2 which seems to be forgotten. Therefore significant modification is needed to correct all the errors.

Regarding experiments, some data-set in Table 1 does not seem to me relevant to the paper (Assumption 1), in particular those with r < 10. Figure 6 shows that the gradient flow is close to a straightly line, suggesting that the gradient descent algorithm follows a convex landscape. The Figure 6(b) seems to me have not converged yet, as at step 10,000, the cosine is not as flat as the others. This means that maybe the gradient flow does not converge to the local minima (U^*,V^*). Further explanation about this is needed in the paper. Regarding optimization efficiency, it is not that convincing since even in the over-parameterized regime: the landscape become more convex, but there can be a lot of local minima which are not as good as the global minima. Therefore from an optimization perspective, finding global minima still remain challenging. I think it would be better to mention this somewhere in the paper.

Minor typo includes:
equation (5), write ||W_2||_F^2.
Equation (12), hat 1 should be 1


**Experience Assessment:**

I have read many papers in this area.

**Review Assessment: Checking Correctness Of Derivations And Theory:**

I carefully checked the derivations and theory.

**Review Assessment: Checking Correctness Of Experiments:**

I assessed the sensibility of the experiments.

**Review Assessment: Thoroughness In Paper Reading:**

I read the paper at least twice and used my best judgement in assessing the paper.

---

> ### Author Response · Authors · 2019-11-12
> **Thanks for the great comments, two questions**
>
> We thank the Reviewer for reading our paper carefully, and for the constructive comments. The proof is long, and we have not been able to compress it. There does indeed seem to be a factor of 2 missing from the definition of w_2, and this would need to be propagated throughout the proof. We share the assessment of the Reviewer that the proof will still be correct with this modification, however, the left-hand side of the equation 3 will grow more than the right-hand side. The Maclaurin series issue is a typo. At any rate, we thank the Reviewer for pointing this out, and pointing out other smaller adjustments needed in the proof.
>
> We are a little unsure about the factor of 2 in fact 10 that the reviewer mentions being forgotten. We assume it is the fact that the mean of the term in equation 3 will change which isn’t mentioned in the text? Or is it the factor from the proposed w_2 -> 2 w_2 change?
>
> Regarding the data sets that are relevant we believe that star convexity is a general phenomenon that occurs for a lot of data sets. We are only able to prove it for some parameters, but we believe that it holds more broadly, and, indeed, that’s what our experiment suggests. We argue that the fact that this property holds more broadly is a strength of the paper.
>
> In figure 6b we could indeed run the algorithm for more steps. We wanted to run all algorithms the same number of steps and there were computational limits regarding running too many steps on the larger data sets. Our theoretical results assume a global minima, but in practice one might indeed converge to local minima. Gradient descent seems to follow a star-convex path towards this local minimum. Indeed, they might not be global minima, but they seem to exhibit nice star-convexity properties nonetheless; this could be clarified in the paper. Is this the type of clarification the reviewer was looking for?
>
> We also thank the Reviewer for pointing out the typos.

---

> > ### Author Response · Authors · 2019-11-12
> > **changes made**
> >
> > * we have replaced w_2 with 2 w_2 in equation 3 and have propagated this change throughout the proof.
> > * the Mclaurin series issue has been fixed, it was just a typo.
> > * mu is indeed the mean of Z. We have made this clear.
> > * “lemma 9” was supposed to be “fact 9”. We have fixed this typo.
> > * we have made a clarification regarding local and global optima.
> > * we have commented on the global vs local minima issue.
> > * we have fixed the typos.

---

> > ### Author Response · Authors · 2019-11-15
> > **changes made**
> >
> > * we've clarified constant 2 from fact 10.

---

### Decision · Program_Chairs · 2019-12-19

**Decision:**

Reject

**Comment:**

The paper derives results for nonnegative-matrix factorization along the lines of recent results on SGD for DNNs, showing that the loss is star-convex towards randomized planted solutions.

Overall, the paper is relatively well written and fairly clear.  The reviewers agree that the theoretical contribution of the paper could be improved (tighten bounds) and that the experiments can be improved as well. In the context of other papers submitted to ICLR I therefore recommend to reject the paper.